# Neuropeptide Y Peptide Family and Cancer: Antitumor Therapeutic Strategies

**DOI:** 10.3390/ijms24129962

**Published:** 2023-06-09

**Authors:** Manuel Lisardo Sánchez, Francisco D. Rodríguez, Rafael Coveñas

**Affiliations:** 1Laboratory of Neuroanatomy of the Peptidergic Systems, Institute of Neurosciences of Castilla and León (INCYL), University of Salamanca, 37008 Salamanca, Spain; lisardosanchez8@gmail.com; 2Department of Biochemistry and Molecular Biology, Faculty of Chemical Sciences, University of Salamanca, 37008 Salamanca, Spain; lario@usal.es; 3Group GIR-USAL: BMD (Bases Moleculares del Desarrollo), University of Salamanca, 37008 Salamanca, Spain

**Keywords:** NPY, PYY, PP, neuropeptide Y, peptide YY, pancreatic polypeptide, NPY receptor antagonist, Y receptor, apoptosis, metastasis, cancer progression

## Abstract

Currently available data on the involvement of neuropeptide Y (NPY), peptide YY (PYY), and pancreatic polypeptide (PP) and their receptors (YRs) in cancer are updated. The structure and dynamics of YRs and their intracellular signaling pathways are also studied. The roles played by these peptides in 22 different cancer types are reviewed (e.g., breast cancer, colorectal cancer, Ewing sarcoma, liver cancer, melanoma, neuroblastoma, pancreatic cancer, pheochromocytoma, and prostate cancer). YRs could be used as cancer diagnostic markers and therapeutic targets. A high Y1R expression has been correlated with lymph node metastasis, advanced stages, and perineural invasion; an increased Y5R expression with survival and tumor growth; and a high serum NPY level with relapse, metastasis, and poor survival. YRs mediate tumor cell proliferation, migration, invasion, metastasis, and angiogenesis; YR antagonists block the previous actions and promote the death of cancer cells. NPY favors tumor cell growth, migration, and metastasis and promotes angiogenesis in some tumors (e.g., breast cancer, colorectal cancer, neuroblastoma, pancreatic cancer), whereas in others it exerts an antitumor effect (e.g., cholangiocarcinoma, Ewing sarcoma, liver cancer). PYY or its fragments block tumor cell growth, migration, and invasion in breast, colorectal, esophageal, liver, pancreatic, and prostate cancer. Current data show the peptidergic system’s high potential for cancer diagnosis, treatment, and support using Y2R/Y5R antagonists and NPY or PYY agonists as promising antitumor therapeutic strategies. Some important research lines to be developed in the future will also be suggested.

## 1. Introduction

Chemotherapy and radiotherapy are, unfortunately, unspecific antitumor treatments currently used in clinical practice. Both treatments lead to severe side effects (e.g., radiotherapy can damage healthy tissues); thus, new antitumor strategies are urgently needed. Peptidergic systems are involved in cancer progression and regulate crucial roles such as cell proliferation, migration, and angiogenesis [1,2]. In this sense, the knowledge of the roles played by the substance P/neurokinin-1 receptor system in cancer progression has been dramatically increased in recent years and, according to critical findings, the repurposing of aprepitant (a neurokinin-1 receptor antagonist currently used in clinical practice as an antiemetic) as an antitumor agent has been proposed [2,3]. One essential difference between normal and cancer cells is the expression of peptide receptors [2,4]. Compared with normal cells, these receptors are generally overexpressed in cancer cells, and this crucial observation opens the door to developing specific antitumor strategies alone (e.g., using peptide receptor antagonists) or combined with chemotherapy or radiotherapy [2,4,5,6]. Accordingly, peptide–drug conjugates can be used as a targeted antitumor therapy against cancer cells overexpressing peptide receptors [4]. This therapeutic strategy could avoid the administration of compounds showing no selectivity against tumor cells, as unfortunately occurs with chemotherapeutic agents.

On the contrary, due to the overexpression of peptide receptors and through targeted antitumor therapy (peptide–drug conjugates), selective drug delivery (e.g., cytotoxic molecules, radionuclides) can be achieved [4,6]. A tumor-to-normal peptide receptor expression ratio of 3/1 or higher is required to achieve selective drug delivery into tumors, whereas healthy tissues are spared [4]. The peptide–drug conjugate strategy is used for cancer diagnosis and treatment. For example, primary tumors and metastatic sites were observed in breast cancer patients treated with a technetium-99 m labeled [F7, P34]-NPY (neuropeptide Y) conjugate, but no peptide uptake was observed in healthy individuals [7]. This finding shows the high specificity of the peptide–drug conjugate strategy; hence, this research line must be potentiated and developed in the future.

Another crucial research line must be thoroughly investigated: using peptide receptor antagonists as antitumor drugs [2]. These antagonists are known to block tumor cell proliferation and migration, angiogenesis, and promote the death of tumor cells by apoptosis [2,8,9]. This is a more straightforward antitumor strategy than using peptide–drug conjugates because, to synthesize these conjugates, the cytotoxic agents must be linked to the peptides. In addition, the administration of NPY receptor antagonists showed no severe side effects in both experimental animals and humans [10]. Accordingly, the main aim of this review is to update the currently available information on the involvement of NPY in cancer and suggest the use of NPY receptor antagonists as a promising antitumor strategy against tumor cells overexpressing NPY receptors (YRs). Moreover, the involvement of pancreatic polypeptide (PP) and peptide YY (PYY), members of the NPY peptide family and less studied in cancer, will also be reviewed.

## 2. The Neuropeptide Y System: Peptides and Receptors

The NPY family in mammals forms a system of three homologous peptides with diverse functions: NPY, expressed in the brain and gut, and PYY and PP, only present in the digestive system [11]. These hormones bind to and activate a group of YRs [12,13], building a complex gut–brain molecular network influencing numerous physiological events related to food intake, energy balance, glycemia homeostasis, pain, bone metabolism, cardiovascular function, stress, anxiety, and cell growth and proliferation [14,15,16,17,18].

### 2.1. The Neuropeptide Y and Its Coding Gene

NPY is a neurotransmitter or modulator with a sequence of 36 amino acids determined approximately three decades ago [19]. Dipeptidyl peptidase-4 (DPP-4) hydrolyzes NPY to generate its N-terminal truncated NPY3-36 product, a selective Y2R agonist [20]. The polypeptide appearance extends through the brain–gut axis (enteric neurons, primary afferent neurons, and several brain areas and neuronal pathways) [21] and exerts biological activity by binding to all YRs in human cells.

The human gene h*NPY* on chromosome 7p15.3 encodes a pro-NPY polypeptide containing the 36 amino acid sequence corresponding to NPY (Figure 1). Posttranslational modification of pro-NPY consists of a hydrolytic cleavage to separate the N-terminal signal peptide and the C-flanking peptide and the amidation of the C-terminal tyrosine to liberate the mature sequence of NPY [22]. Among the gene variants described for the *NPY* gene, the substitution of leucine in position 7 of the signal peptide to proline (variant rs780799208) is associated with high alcohol consumption and cardiovascular pathologies [22,23].

### 2.2. The Peptides Peptide Tyrosine Tyrosine (PYY) and Pancreatic Polypeptide (PP) and Their Coding Genes

Enteroendocrine nutrient-sensing L-cells in the ileum and colon secrete polypeptide PYY1-36 after meals [27]. It is also produced in the central nervous system [18,28]. The 36 amino acid sequence of PYY was first determined in PYY isolated from the porcine intestine [19]; following hydrolysis by dipeptidyl peptidase-4 (DPP-4), its product is the N-terminal truncated PYY3-36 [29]. Both peptides circulate in the bloodstream to participate in the gut–brain axis controlling many different physiological processes. The complete form of PYY has an affinity for all human Y1R, Y2R, Y4R, and Y5Rs, whereas PYY3-36 exhibits high specificity for Y2R [30]. Degradation products of PYY peptides after cleavage of C-terminal residues are under study to determine possible metabolic roles [18,31,32,33].

The endocrine pancreas secretes pancreatic polypeptide (PP), a hormone involved in the gut–hypothalamic axis control of satiety and appetite [34]. It exhibits a selective affinity for Y4R [12]. PP is the first identified and characterized peptide of the NPY family [35]. Both peptidases DPP-4 and neprilysin (NEP) rapidly degrade the molecule, which has a short life once released into the bloodstream [36].

The human gene h*PYY* on chromosome 17 (17q21.1) encodes two functional peptides, PYY (36 amino acids) and the amino-truncated shorter form PYY3-36 (34 amino acids) [37,38], after proteolytic processing of the encoded 97 amino acid propeptide [22] (Figure 2). Both PYY peptides, NPY and PP, share high sequence identity and an identical pentapeptide with an amidated tyrosine at the C-terminal end.

On chromosome 17, around 10 kb apart from the h*PYY* gene, the h*PPY*, a tandem duplication of the h*PYY* gene, encodes the pancreatic polypeptide (PP) and pancreatic icosapeptide [11,39,40] (Figure 2).

**Figure 2 ijms-24-09962-f002:**
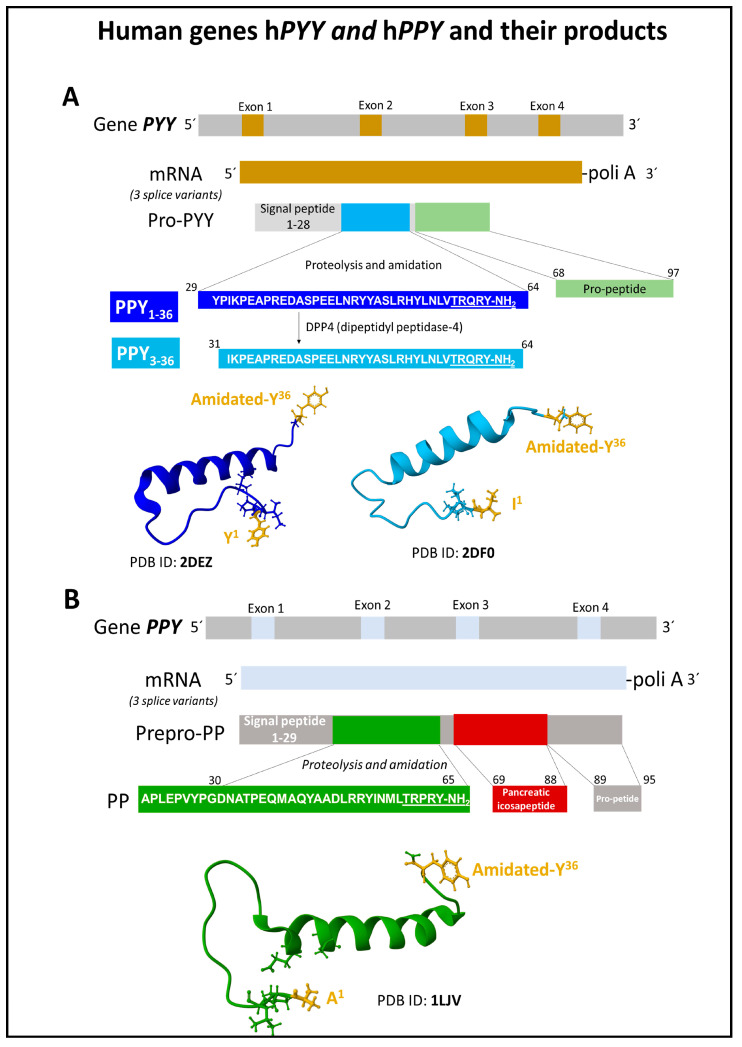
Human gene h*PYY* and its main encoded products, PYY and PYY3-36 (**A**), and human gene h*PPY* and its main encoded products, PP and icosapeptide (**B**). The numbers indicate amino acid positions within the peptides. The NMR secondary structures of PPY (2DEZ, [41]), PYY3-36 (2DF0, [41]), and PP (1LJV, [42]) obtained from the Protein Data Bank [25] were drawn with the free web-based software Mol* (https://molstar.org/ (accessed on 5 May 2023)) [26]. The underlined C-terminal pentapeptide sequence is identical in all three members of the human NPY family.

### 2.3. The Structure and Dynamics of the Neuropeptide Y Receptors (YR) 

The NPY polypeptides and their receptors build a piece of complex molecular machinery. They all have a 36 amino acid sequence length and an amidated C-terminal tyrosine (Figure 1 and Figure 2). A commonly denominated PP-fold (hairpin structure) tertiary structure consisting of an N-terminal extended polyproline-like helix and a C-terminal alpha helix in PPY and PP delineating a hydrophobic pocket determines their binding to specific Y receptors and their consequent bioactivity [43,44,45]. In contrast, the structure of NPY slightly differs from the other two members since its N-terminal domain appears disordered (Figure 1) [43]. In humans, four functional Y receptor types, Y1R, Y2R, Y4R, and Y5R, broadly distributed in the central, peripheric nervous system and other tissues, process the signaling of this group of peptides. The affinities and potencies of the NPY family peptides are different for the four mentioned receptors [12]. This family of receptors controls food intake and obesity, stress, anxiety, and cancer development [1,46,47,48]. We next describe the principal features of the architecture, dynamics, and signaling of this family of human receptors belonging to the G-protein-coupled receptors (GPCR), type A.

#### 2.3.1. Y1R 

The neuropeptide Y receptor 1 (Y1R) is a 384 amino acid membrane protein with affinity for the main NPY peptides in the following rank order: NPY > PYY > PYY3-36 > PP. Posttranslational modifications of the protein include glycosylation of Asn 2, 11, and 17, a disulfide bridge between Cys113 and Cys198, a lipidation site on Cys338, and a phosphorylated Ser368 (UniProt, P25929 [22,49]). Its sequence extends through the plasma membrane with seven transmembrane domains (Figure 3).

Structural analysis of Y1R bound to NPY and a G-protein (active conformation) by cryo-electron microscopy [50,51] and of Y1R antagonist UR-MK299 (inactive conformation) by X-ray diffraction [16] has provided detailed information on how the ligands interact with the receptor protein. Analysis of differences in the binding mode of natural agonists and antagonists contributes to better ascertaining the receptor’s molecular dynamics. The functional study of mutations and molecular dynamics simulations also add valuable information concerning the inactive state structure and how the natural agonists activate the receptor by recruiting a G-protein transducer. The C-terminal unstructured tail of NPY enters deeply into the protein and interacts with a pocket conformed by the transmembrane domains TM3, TM4, TM6, and TM7. The amphipathic helix wobble connects with the extracellular domains of Y1R. Given its disordered and flexible structure, the binding coordinates of the N-terminal region of NPY are more challenging to determine [50]. The conformational flexibility of NPY allows the binding to different YRs by adapting its structure to the receptors [51].

Upon binding, NPY forces an outward displacement of TM7, opens the binding site [50], and induces conformational changes consisting of rearrangements of some receptor residues that lead to recruiting a G-protein, triggering receptor activation.

#### 2.3.2. Y2R

The neuropeptide Y receptor 2 (Y2R) is a 381 amino acid membrane protein with affinity for the NPY peptides in the following rank order: PYY > NPY > PYY3-36 > PP. Posttranslational modifications of the protein include glycosylation of Asn 11, a disulfide bridge between Cys123 and Cys203, a lipidation site on Cys342, and a phosphorylated Ser at 251, 351, 369, and 374 positions (UniProt, P49146 [22,49]). Its sequence extends through the plasma membrane with seven transmembrane domains (Figure 4).

The structure of an engineered crystal of Y2R copurified with the specific antagonist JNJ-31020028 [51] provided conformational and functional data indicating that the binding of the antagonist forces inward and outward movements of transmembrane domains II and VI and suggests that the complex Y2R-JNJ-31020028 adopts an inactive state conformation similar to the idle state described for Y1R [16].

Y2R with two mutated positions, namely H1943.51Y and S2806.47C (superscripts depict Ballesteros–Weinstein numbers), was complexed to natural ligands NPY and PYY3-36 to analyze its structure by cryo-electron microscopy. Additional molecular dynamics calculations, functional studies, and bioluminescence resonance energy transfer (BRET) studies revealed important details concerning unique features of the binding of peptide PYY3-36, receptor activation states, and coupling mechanisms to transducers [53]. Peptide binding to Y2R disturbs a polar interaction between Q1303.32 and H3117.39, establishing hydrogen bonds with the ligand amidated Tyr36. These movements displace other amino acid residues and finally conform the ligand pocket within Y2R. Cryo-electron maps of NPY bound to Y2R coupled to heterotrimeric Gi protein show that the C-terminal pentapeptide inserts deep into the Y2R structure, forcing the α-helix to incline toward the N-terminal domain establishing contact with extracellular domains 2 and 3. The N-terminal region accommodates the outer part of receptor helix 5 [51]. Additionally, toggle residue W6.48 interacting with the ligand C-terminal residue Y36 seems to influence the Y2R-dependent activation of transducers Gi and β-arrestin [54]. These studies’ principal structural dynamics results are essential to ascertain the molecular basis of ligand–receptor interactions applied to decipher the pathophysiological role fully and to rational and successful drug design.

#### 2.3.3. Y4R

The neuropeptide Y receptor 4 (Y4R) is a 375 amino acid membrane protein with affinity for the NPY peptides in the following rank order: PP > PYY > NPY > PYY3-36. Posttranslational modifications of the protein include glycosylation of Asn 2, 19, 29, and 187, a disulfide bridge between Cys114 and Cys201, and a lipidation site on Cys340 (UniProt, P50391, [22,49]). Its amino acid sequence extends through the plasma membrane with seven transmembrane domains (Figure 5).

PP prefers Y4R over other YRs. Calculating binding energies from molecular dynamics simulation and docking analysis showed that PYY binds Y4R with a higher affinity than Y2R [57]. Based on the structure of PP, synthesized short peptides with modifications and amino acid substitutions have provided relevant information concerning the binding mode and activity. One fact is that the C-terminal amidated end is essential for binding and activity [58]. Short-cycled peptides with carbamoylated arginines or arginines substituted with N-acylated ornithine exhibit better affinities and selectivities for Y4R and serve as a reference for better defining binding sites and compounds with agonist, partial agonist, and antagonist properties [59].

Cryo-electron microscopy studies of the complex formed by Y4R-Gi heterotrimeric protein-PP report the extensive peptide site delimited by the extracellular loops and transmembrane domains 2 to 7. The carboxy end of PP (Y36-T32) penetrates deep within the packed receptor structure [55] (Figure 5B). 

The stereoisomer (S)-VU0637120 is a synthetic antagonist that selectively inhibits Y4R activation by a negative allosteric mechanism [56]. The compound occupies a site within the receptor structure defined by specific principal interactions with amino acids in transmembrane domains 1, 2, 3, 4, and 7 (Figure 5D). Additionally, the extracellular loop 2 (ECL2) plays a vital role in the allosteric regulation performed by the antagonist. Both binding sites slightly overlap according to docking and mutagenesis studies [56]. The definition of orthostatic and allosteric sites within Y4R offers vital information for designing selective allosteric agonists and modulators, permitting fine-tuned regulation of altered Y4R activity.

#### 2.3.4. Y5R

The neuropeptide Y receptor 5 (Y5R) has 445 amino acids. The affinity of Y peptides for Y5R is higher for NPY, followed by PYY and PP (NPY > PYY > PP) [60]. Posttranslational modifications of the protein include glycosylation of Asn 10 and 17, a disulfide bridge between Cys114 and Cys198, and a lipidation site on Cys442 (UniProt, Q15761, [22,49]. Its sequence extends through the plasma membrane with seven transmembrane domains (Figure 6).

The AlphaFold method [61] allows the prediction of a protein structure with the serpentine arrangement disposed within the plasma membrane [22]. Additionally, a structural model of the interaction of NPY with Y5R based on computational and biochemical analysis places the peptide close to the ECL3, with the alpha-helical domain wrapping the ECL1. The peptide is secured through a hydrophobic grouping formed by L4.69, L5.24, L24, and I28 [62]. Mutagenesis studies provided fundamental ligand–receptor interactions with R25 and D2.68 and R33 with D6.59 [63]. Further structural determination of Y5R by X-ray diffraction, NMR, or cryo-electron microscopy will permit us to better define the binding site for agonists, antagonists, and allosteric modulators, as well as the molecular movements responsible for its activation or dormant states.

### 2.4. Intracellular Signaling of Y Receptors

The functional diversity of this multi-receptor multiligand system comes from tissue localization and abundance, together with receptor structural features that adapt to the natural ligands in different ways and determine their signaling profile. It also depends on the Y receptors’ biased signaling mechanisms when triggered by the NPY family of peptides and the cross-talk with other receptor systems [53,64,65].

Peptide binding activates YRs, which, through differentiated transduction mechanisms (Figure 7), trigger biochemical events leading to metabolic changes, adaptations, and gene expression modulations, eventually contributing to functional cell metabolism homeostasis, cell growth, motility, and migration [13,66,67,68,69].

Human YRs activate transducers, heterotrimeric G proteins, and β-arrestins, resulting in diversified intracellular signaling pathways. YRs 1, 2, and 5 preferentially turn on Gi/o alpha subunits and inhibit cyclic adenosine monophosphate (cAMP) formation. Gi/o may also regulate the activity of plasma membrane K^+^ and Ca^2+^ ion channels. Y2R and Y4R stimulate Gq/11 alpha subunit dissociation to increase intracellular calcium levels and protein kinase C (PKC) activation [60]. YR signaling routes may coexist and connect with other receptor-triggered pathways turning on downstream biochemical signaling changes and leading to adaptive changes controlling cell behavior [70,71,72,73,74,75,76,77] (Figure 7).

Determining and quantifying the conformational landscape of YR impacts activating defined signaling events is capital to designing new drugs that may interfere with excessive functional or non-functional receptor states and control their role in biochemical mechanisms leading to disease [78]. Additionally, analyzing the mechanisms underlying receptor activation and desensitization will aid the chemical design of effective drugs to control YR biochemical behavior [79].

## 3. Involvement of Neuropeptide Y, Peptide YY, and Pancreatic Polypeptide in Cancer

NPY has been negatively correlated with COL5A1, COL3A1, and COL4A1 collagen gene expressions in a pan-cancer study [80]. NPY promoted an antinociceptive effect that was inhibited with Y1R or Y2R antagonists (BIBO3304 and BIIEo246, respectively), and the peptide was upregulated in cancer-induced bone pain [81]. NPY induced inflammation-induced tumorigenesis by promoting epithelial cell proliferation (intestinal epithelial T84 cell line) [82], and the peptide, through the phosphatidyl-inositol-3-kinase (PI3K)-β-catenin signaling, was involved in this proliferation, decreased apoptosis and p21 expression, increased c-myc/cyclin D1 expressions and inhibited miR-375 expression (apoptosis regulating microRNA) [82]. NPY also favored, through the p38 mitogen-activated protein kinases (MAPK) and extracellular signal-regulated protein kinase (ERK), the migration of human monocyte-derived immature dendritic cells as well as their transendothelial migration [83]. Y1R mediated these actions. However, by upregulating interleukins 6 and 10, NPY promoted a T helper 2 polarizing profile to dendritic cells [83]. This fact means that NPY, via Y1R, can exert a pro-inflammatory action (immature dendritic cell recruitment) or an anti-inflammatory effect (favoring a T helper 2 polarization). NPY also linked longevity and dietary restriction in experimental animal models, and spontaneous tumors were attenuated in NPY-null animals [84]. This study showed that NPY, through dietary restriction, exerted an important role in cancer suppression and lifespan extension. However, another study showed that tumor development was neither reduced nor delayed by calorie restriction in NPY knockdown animals, which were used as an appetite suppression model [85]. These data are only a few examples demonstrating the involvement of NPY in cancer. In this section, the participation of NPY, PYY, and PP in the development of many cancer types is reviewed.

### 3.1. Brain Tumors

A high expression of Y1R has been reported in brain tumors, including glioma [86]; however, in patients with glioblastoma, no plasma level alteration was observed during the initial clinical manifestations of the disease [87]. Glioblastoma (grade IV) showed a high expression of Y2R, and intratumoral nerve fibers containing NPY were observed [88]. Therefore, the peptide can control tumor development after release from nerve fibers. Medulloblastomas, meningiomas, and primitive neuroectodermal tumors of the central nervous system also expressed Y1R and Y2R, whereas astrocytomas (grades I to III) showed Y2R [88]. Higher NPY levels were observed in differentiated tumors (grade I/II astrocytomas) than in poorly differentiated neurological tumors [89]. Finally, glutamate increased the expression of NPY in rat C6 glioma cells [90], and downregulation of NPY mRNA expression was observed when these cells were implanted into the rat’s third ventricle [91].

The presence of PYY has been reported in cerebellar hemangioblastomas [92]. PYY was observed in the perinuclear region of stromal cells placed in small clusters and scattered and distributed by the tumor parenchyma.

### 3.2. Breast Cancer

NPY promoted the proliferation and migration of breast cancer cells and angiogenesis [93,94]. Y1R, Y2R, and Y5R expressions have been reported in breast cancer cell lines and tumor tissues, and tumor cell growth and migration were mediated by Y5R [10,94]. An overexpression of Y1R has been observed in primary human breast tumors (>85% of the studied samples) and in breast cancer-derived metastases (100%), whereas normal human breast tissues mainly expressed Y2R [6]. Thus, the mechanisms involved in the neoplastic transformation change the receptor expression from Y2R to Y1R [95]. Y1R is coupled to cAMP and Ca^++^ pathways in MCF-7 cells (NPY mobilizes intracellular Ca^++^) [96]. NPY blocked forskolin-stimulated cAMP accumulation and promoted ERK phosphorylation and tumor cell migration; both cell growth and migration were inhibited with the Y5R antagonist CGP71683A [97]. This antagonist also induced the death of tumor cells expressing Y5R [97]. The proliferative effect was mediated through an increased ERK1/2 phosphorylation mechanism [94] and Y5R-mediated cAMP inhibition in breast cancer cells (BT-549) [97]. Vascular endothelial growth factor (VEGF) levels increased in breast cancer cells (4T1) treated with Y5R agonists but not when these cells were treated with Y1R or Y2R agonists [93]. Y5R agonists also promoted VEGF release from breast cancer cells favoring angiogenic mechanisms [93]. 

A higher *Y1R* gene expression predicts a better relapse-free survival/overall survival in estrogen receptor-positive breast cancer patients [98]. Estrogens upregulate Y1R receptors, which control estrogen-induced tumor cell proliferation [96]. The inhibitory action exerted by NPY on estradiol-stimulated growth of estrogen receptor-positive breast cancer cells was mediated by Y1R, and its expression could be used as a biomarker for survival/endocrine sensitivity in estrogen receptor-positive breast cancer patients [98]. Serum Y1R has also been suggested as a biomarker for metastasis/prognosis in patients with breast cancer expressing a high Y1R level [99]. This high level has been positively correlated with lymph node metastasis/clinical stage, and, in addition, breast cancer patients with Y1R expression showed a shorter cancer-specific survival than those patients without Y1R expression [100]. A higher Y1R expression has been reported in malignant tissues of patients with breast cancer than in non-neoplastic tissues: this high Y1R expression was associated with metastasis, advanced stages, poor Nottingham prognostic index, and perineural invasion [101]. NPY regulated breast cancer development and osteoporosis progression since the osteogenic response was mediated by Y1R/Y2R [102]. Hence, peptides and YRs could be used to diagnose and develop new therapeutic strategies for breast cancer and osteoporosis.

The pre-pro NPY protein expression was upregulated in human breast adenocarcinomas, but the NPY protein level was downregulated [103]. This observation suggests that the involvement of the proprotein convertase 2-mediating processing in the previous finding is weak. Y1R expression was upregulated by 17β-estradiol and downregulated by the anti-estrogen agent fulvestrant in breast cancer cells (MCF-7), whereas tamoxifen induced the total loss of Y1R in MCF-7 xenografts [104]. Supplementation with decaffeinated green coffee extract in breast cancer survivors with obesity did not affect serum NPY levels [105]. However, social isolation increased the risk of cancer development and NPY levels [106], and in an experimental model of breast cancer, chronic stress increased the extent/frequency of bone metastases [107].

The expression of membrane proteins can be changed due to multidrug resistance. Multidrug-resistant breast cancer cells (MCF-7/ADR) overexpress Y1R and P-glycoprotein [108]. A selective Y1R ligand (Asn6, Pro34-NPY) co-administered with tariquidar (a P-glycoprotein inhibitor) exerted a synergistic effect and improved the therapeutic efficacy against multidrug-resistant breast cancer cells [108]. This co-administration strategy improved the delivery of tariquidar and the chemotherapeutic drug doxorubicin into breast cancer cells, inhibiting tumor cell growth. NPY-loaded immune stimulating complexes (ISCN formulation) decreased growth and cytokine levels in MCF-7 tumor cells; cells were blocked in the G0/G1 phase, and many suffered apoptosis after treatment with the ISCN formulation [99]. Similar results were found after using NPY-decorated gold nanoclusters to treat MCF-7 cells [109].

PYY blocked the growth of breast tumor cells (e.g., MCF-7, ZR-75), and the same effect was observed in gastric tumor cell lines [110,111,112]. Compared to when administered alone, an increased antitumor action was observed when PYY and vitamin E were co-administered [111]. PYY also inhibited the growth of breast cancer cells (MCF-7) in vivo, and cAMP levels were reduced in these cells after treatment with PYY [110].

### 3.3. Cholangiocarcinoma 

NPY blocks tumor cell growth and invasion in cholangiocarcinoma; both effects were counteracted with anti-NPY antibodies [113]. These effects were also inhibited with Y2R inhibitors but not with Y1R/Y5R inhibitors and were associated with PKCα activation and an intracellular d-myo-inositol 1, 4, 5-triphosphate increase [113]. Moreover, NPY immunoreactivity was higher in the center of tumors than in the invasion front [113].

### 3.4. Colorectal Cancer 

Plasma NPY concentration is decreased in patients with colorectal or gastric carcinomas, and this decrease was associated with tumor size (>5 cm) and body weight loss (>3 kg) [114]. *NPY* gene methylation has been suggested as an early biomarker for metastatic colorectal cancer progression [115,116,117]. In this context, a study has reported that hypermethylation of the *NPY* gene in circulating tumor DNA could be a biomarker for colorectal cancer screening and diagnosis [118]. The latter study also noted that the degree of methylation was not correlated with mutation status, tumor stages, or histological characteristics [118]. More studies reported that plasma methylated circulating tumor DNA could be used as a universal biomarker in metastatic colorectal tumors [119,120] since methylated circulating tumor DNA was detected in all plasma patients studied suffering from this disease [119]. It has also been suggested that plasma NPY methylation analysis would evaluate the clinical benefits of last-line treatment with regorafenib in metastatic colorectal cancer patients [120]. Notably, the median survival for patients with a methylated circulating tumor DNA level above the median was 4.3 months; however, it was 7.6 months when the value was below the median [120].

Y1R has not been detected in colorectal cancer [6]. NPY and Y2R are overexpressed in human colon adenocarcinomas, orthotopic HT29, and VEGF-A-depleted orthotopic HT29 tumors [121]. Y2R antagonists blocked HT29 tumor growth and NPY-induced angiogenesis on colonic endothelial cells expressing Y2R [121]. NPY, released from tumor cells, controlled the activation of the ERK/MAPK pathway in the latter cells. The data suggest that targeting the NPY/Y2R system is a promising antitumor strategy to treat colon adenocarcinomas.

Moreover, in the myenteric/submucous plexuses, a decrease in the number of NPY-immunoreactive neurons/nerve fibers was observed in colorectal tumor samples [122], and the absence of perivascular nerves into the tumors and the loss of perivascular innervation in the submucosa placed close to the tumors have been reported in colorectal cancer [123]. In this study, NPY immunoreactivity loss was progressively higher with advanced tumor stages. Electron microscopy showed that perivascular nerve fibers were absent in colorectal liver metastasis. In addition, blood vessel walls contained endothelial cells but were only surrounded by a complete/incomplete layer of smooth muscle-like cells [124]. Thus, the histological organization of these blood vessels was significantly altered.

Plasma PYY level is not related to colorectal cancer risk [125]. PYY and PP have been observed in rectal tumors [126], and the density of cells expressing PYY in these tumors was small-moderate compared to those cells containing PP [127]. The number of cells containing PP was also higher than those containing PYY in rectal carcinoids [126]. PYY-positive carcinoids of the appendix contain almost exclusively cells expressing the peptide, whereas rectum tumors contain sporadic PYY cells [128]. Moreover, the number of cells containing PYY was meager in jejunal, duodenal, and gastric carcinoids [128]. A study reported that the highest PYY level was observed in tubular adenomatous colonic polyps; this level was lower in villous adenomatous colonic polyps, and the lowest PYY level was reported in colon carcinomas [129]. Thus, low PYY levels indicate the malignant potential of these alterations, and the data show that an adenoma-carcinoma sequence occurs in colonic cancer. In this regard, another study reported that PYY-immunoreactivity was higher in normal colon mucosa than in adenocarcinomatous tissues of any part of the rectum and colon [130].

Moreover, a low PYY expression has been observed in colorectal cancer tissue; the overexpression of this peptide promoted apoptosis in HCT116/HT29 tumor cells and blocked the proliferation, migration, and invasion of these cells, and PYY released from neurons suppressed tumor growth [131]. Cells expressing PYY were increased in an experimental animal model of colonic carcinoma, whereas no cell containing PP was observed [132]. HCT116 and Caco2 colon cancer lines increased Bcl-2 expression (which blocks apoptosis; its overexpression is associated with drug resistance) and counteracted the antitumor effect mediated by PYY [133].

Patients with colon cancer showed a higher level of PP than that observed in patients with rectal tumors or healthy individuals [134]. PP, located in the cytoplasm, was also observed in rectal neuroendocrine tumors [135]. Tumor cells release PP. This has been demonstrated in gangliocytic paraganglioma tumors (a benign neuroendocrine tumor generally located in the duodenum), and PP was detected in the plasma of patients suffering from this disease [136].

### 3.5. Esophageal Cancer

PYY3-36 blocked the growth of Barrett’s esophageal cancer cell lines (BIC, SEG-1) [137]. Apoptotic mechanisms were observed after PYY3-36 treatment; for example, the peptide augmented the late apoptotic activity in SEG-1 cells [137].

### 3.6. Ewing Sarcoma

The EWS-FLI1 fusion protein is an aberrant transcription agent that upregulates genes in Ewing sarcoma (e.g., NPY, Y1R, Y5R) [138] and triggers the malignant transformation in Ewing sarcoma [10]. Ewing sarcoma intratumoral blood vessels expressed Y1R [139]; Ewing sarcoma cells express NPY, Y1R, and Y5R, and the peptide via both receptors promoted the death of tumor cells, which was mediated by poly(ADP-ribose polymerase) (PARP-1) and apoptosis-inducing factor (AIF) [10,138,139,140]. It has been suggested that NPY/Y1R/Y5R are involved in maintaining a highly undifferentiated phenotype in Ewing sarcoma because both receptors mediated pluripotency and human embryonic stem cell self-renewal [141]. NPY blocked, via Gi-coupled-Y1R, cAMP formation in Ewing sarcoma cells [142], and the peptide, through the inhibitory guanine nucleotide regulatory protein of adenylate cyclase, also blocked the dopamine-induced cAMP formation in Ewing sarcoma cells (WE-68), but not basal cAMP formation [143]. However, under hypoxia conditions, NPY acted as a growth-promoting agent because these conditions favored the expression of NPY/Y2R and increased the level of dipeptidyl peptidase IV, which cleaved NPY, originating the NPY3-36 fragment, which acted as a selective Y2R/Y5R agonist [10,140]. NPY3-36 does not bind to Y1R. It does not favor the death of Ewing sarcoma tumor cells, and the blockade of the activity of dipeptidyl peptidases decreased tumor cell survival, which was attenuated with Y1R/Y5R antagonists [138,140]. Thus, hypoxia counteracts the growth-inhibitory pathway mediated by Y1R/Y5R and activates the Y2R/Y5R/dipeptidyl peptidase IV/NPY3-36 pathway, which mediates a growth-promoting action. Moreover, NPY3-36 also favored the migration of tumor cells and angiogenesis [10,140], and intratumoral endothelial cells showed Y2R, expressed under hypoxia conditions, and mediated the proliferative action promoted by NPY [10]. Accordingly, Y2R antagonists decreased tumor vascularization [10]. NPY/Y5R was highly upregulated in distant metastasis in an experimental model of Ewing sarcoma when compared with the levels reported in primary tumors, and Y2R expression was high in tissues derived from local relapses, suggesting that this receptor is involved in cancer cell invasiveness [144]. Tumor hypoxia exacerbates bone metastasis in Ewing sarcoma; this process is mediated by the hypoxia-induced activation of the NPY/Y5R system, leading to RhoA overstimulation, cytokinesis failure and the formation of polyploid Ewing sarcoma cells [144,145]. The latter cells show a high chromosomal instability, bone invasion capacity, and chemotherapy resistance; however, the blockade of Y5R counteracts bone metastasis and polyploidization [145]. Bone destruction degree in Ewing sarcoma has been positively correlated with the level of NPY release from tumors [144]. Circulating NPY level is high in patients with Ewing sarcoma, and a higher level of the peptide was observed in patients with Ewing sarcoma showing tumors with a pelvic/bone origin [146]. 

### 3.7. Gastric Cancer 

In an experimental model of animals genetically predisposed to develop gastric neuroendocrine neoplasias, the ablation of gastric parietal cells could promote and accelerate the formation of tumors [147]. High serum/tumor tissue PYY levels were reported in these animals; however, it is currently unknown whether PYY is involved in tumor formation or development.

A high plasma PP level was reported in a patient suffering from a gastric tumor [148].

### 3.8. Hemangioma

Y1R, but not Y2R, has been observed in B/T lymphocytes expressing CD45 and mast cells expressing tryptase in infantile hemangiomas, whereas NPY was also observed in those cells expressing Y1R [149].

### 3.9. Head and Neck Cancer

The promoter methylation status of the *NPY* gene has been suggested as an epigenetic biomarker for head and neck cancer prognosis and risk [150].

### 3.10. Kidney Cancer

Y1R expression has been reported in renal carcinomas and Y1R/Y2R expressions in nephroblastomas [151]. A high Y1R density was also observed in the intratumoral blood vessels, and nerve fibers containing NPY were placed close to these vessels and tumor cells [151]. NPY, through Y1R, protected the kidney against cisplatin-induced nephrotoxicity by regulating the p53-dependent apoptotic pathway and decreasing the expression of pro-apoptotic proteins [152].

PP has been reported in cancer cells in a kidney primary neuroendocrine carcinoma [153].

### 3.11. Leukemia

Plasma NPY and NPY mRNA levels in the bone marrow of children with acute leukemia have been studied [154]. A high NPY mRNA level was observed in the bone marrow of children with B-cell precursor leukemia, but no NPY mRNA was reported in children with T-cell leukemia [154]. High plasma NPY levels were only reported in children with B-cell precursor leukemia since these levels were normal in children with myeloid, T-cell, or B-cell leukemias [154]. Plasma NPY levels were higher in children with leukemia than those found in healthy individuals, and these plasma levels were also higher in children with an excellent clinical risk classification [154]. Children with an elevated plasma NPY level had a better outcome than those with a normal level. 

NPY, through Y1R, pertussis toxin-sensitive G proteins, and the PI3K pathway, induced the activation of ERK in human erythroleukemia cells [155]. However, p38 MAPK, jun N-terminal kinase, protein kinase C or phospholipase D were not activated in erythroleukemia cells by NPY [155]. The desensitization of YRs was regulated by a PKC-independent pathway in these cells [156]. NPY promoted the intracellular Ca^++^ release from an inositol 1, 2, 5-triphosphate-sensitive Ca^++^ pool, restored by external Ca^++^ in human erythroleukemia cells [156]. NPY13-36 increased the concentration of intracellular Ca^++^ poorly, and, in addition, this fragment blocked the intracellular Ca^++^ increase mediated by NPY [157]. 

Plasma PYY levels have been measured during chemotherapy treatment in children with acute lymphoblastic leukemia [158]. These levels were high at diagnosis, increased after the induction-consolidation phase, and returned to pretreatment levels after the sixth cycle of chemotherapy [158].

### 3.12. Liver Cancer

NPY, via Y1R, blocks tumor cell growth by inactivating the MAPK pathway in human hepatocellular carcinoma [159]. In this disease, protein/mRNA Y1R levels were decreased; a low receptor expression was associated with poor prognosis, and knockdown of Y1R increased tumor cell proliferation in vitro and tumor growth in vivo [159]. This finding indicates that Y1R overexpression could mediate the blockade of tumor cell proliferation in hepatocellular carcinomas. By contrast, an increased Y5R expression was correlated with survival and tumor growth in hepatocellular carcinoma; NPY/Y5R are involved in tumor cell proliferation, migration, and invasion [160]. However, in the latter study, Y1R/Y2R were downregulated and did not exert a cancer-promoting effect in hepatocellular carcinoma [160]. NPY released from peritumoral hepatocytes favored, through Y5R, tumor progression; tumor growth factor (TGF)-β1 increased NPY expression in hepatocytes and promoted Y5R expression in invasive tumor cells, and the cleavage of NPY by dipeptidyl peptidase IV, upregulated in hepatocellular carcinoma, increased Y5R activation and function [160]. Moreover, the actions favored by TGF-β1 (NPY/Y5R expressions) were mediated by the canonical TGF-β receptor 1, and NPY, through Y5R, activated the MAPK/ERK pathway [160]. The data suggest that Y1R and Y5R play opposite roles in tumor development. NPY and Y2R, but not Y5R expressions, increased in hepatocytes placed close to the stromal–parenchymal interface in hepatocellular carcinoma [161]. In such an interface, NPY can be cleaved by FAP (fibroblast activation protein), a dipeptidyl peptidase upregulated in tumor-activated mesenchymal cells in the stroma. 

NPY decreased PD-1+ T cells and PD-1 expression/cell in T cells of both healthy individuals and patients with hepatocellular carcinoma; the peptide augmented T cell proliferation and the eradication of hepatocellular carcinoma cells [162]. In an experimental animal model of hepatocellular carcinoma, the Ciji-Hua’ai-Baosheng II formula (a traditional Chinese medicine strategy targeting the adverse effects mediated by chemotherapy in cancer patients) increased serum NPY level and NPY expression in the hypothalamus [163]. 

PYY blocked human HepG2 hepatic carcinoma cell growth in an experimental animal model; PYY decreased tumor volume/weight and cAMP level [164]. 

### 3.13. Lung Cancer

*NPY* gene expression is upregulated in patients with lung adenocarcinoma and TP53 mutation [165]. Y1R has not been detected in non-small lung cancer cells [6].

PYY level was higher in patients with lung carcinoma than in healthy individuals [166].

Serum PP level was high in patients with metastatic bronchial carcinoid tumors [167]. PP level was higher in patients with advanced lung cancer than in healthy individuals [168], and in patients with lung carcinomas, PP expression was higher than that reported in healthy individuals [166,169].

### 3.14. Melanoma

NPY expression has been reported in cutaneous melanoma; this expression was higher in melanomas than in melanocytic nevi, and nodular melanomas showed the highest median percentage of tumor cells expressing NPY, followed by superficial spreading melanomas [170]. However, melanoma metastases and lentigo malign melanomas did not express NPY [170]. A high NPY expression has been associated with invasiveness independently of proliferative markers (e.g., mitotic index, ulceration, thickness) [170]. 

The involvement of Y2R in tumor angiogenesis has been studied in normal and obese mice in which B16F10 melanoma cells were administered [171]. Tumor weight decreased in animals treated with the Y2R antagonist BIIE0246. No effect was observed in control mice [171]. Moreover, angiogenesis and serum VEGF level dropped in the obese group treated with the mentioned Y2R antagonist, whereas in this group, no alteration was observed in serum VEGF receptor 1 and nitric oxide [171].

Chemical sympathectomy promotes changes in the tumor microenvironment and decreases melanoma growth [172]. In the tumors of sympathectomized animals, *NPY* gene expression and hypoxic (hypoxia-inducible factor (HIF)1α) and apoptotic (caspase 3, Bcl-2) factors were increased. *Y6R* gene expression has been associated with tumor development in uveal melanoma and suggested as a prognostic biomarker for this disease [173]. A low NPY expression has been associated with scarce E-cadherin expression, high peritumoral mast cell infiltrates, and high cell proliferation in primary cutaneous melanomas [174]. By contrast, a high NPY expression was associated with a better outcome, the presence of adhesion proteins, a decrease in the number of peritumoral mast cells, and better prognostic histological parameters. 

### 3.15. Neuroblastoma

NPY and Y2R are expressed in neuroblastoma cells, and NPY, released from these cells, favored cancer cell proliferation and angiogenesis in an autocrine manner [175]. NPY, through Y2R, increased intracellular Ca^++^ without soliciting Ca^++^ influx in neuroblastoma cells (LN319), which exclusively express Y2R [176]. NPY was also reported in ganglioneuroblastomas [177]; Y1R, Y4R, and Y5R were expressed in SK-N-MC neuroblastoma cells [178], and Y5R mediated their proliferation [179]. The synthesis and release of NPY from neuroblastoma cells are stimulated by the PKC-coupled M3 muscarinic receptor [180,181]. After binding to estrogen receptors alpha, 17 beta-estradiol favored *Y1R* gene transcription [182]. Moreover, retinoic acid decreased *NPY* gene expression in neuroblastoma cells (SH-SY5Y) and benefited the proNPY processing to NPY [183], whereas this acid reduced the expression of YRs in the SK-N-MC neuroepithelioma cell line [184]. NPY, released from neuroblastoma cells, exerted through Y2R an autocrine action on cancer cells favoring cell proliferation [10]. BIIE0246, a Y2R antagonist, inhibited p44/42 MAPK activation, decreasing cell proliferation and promoting apoptosis (mediated by Bim); similar effects were observed with NPY/Y2R small interfering RNA [175]. Moreover, the antagonist exerted an anti-angiogenic action, decreasing the proliferation of endothelial cells (which express Y2R) and hence vascularization; Y2R antagonists, but not Y5R antagonists, decreased this vascularization [10,175,179]. Notably, the proliferative action mediated by Y2R in tumor cells was enhanced when neuroblastoma cells also expressed Y5R [179,185]. Y2R promotes glycolysis in neuroblastoma cells; this process is vital for cancer cells to obtain ATP under hypoxia conditions [186].

NPY expression is increased in neuroblastoma cells treated with valproate [187]; brain-derived neurotrophic factor (BDNF) promoted NPY/Y5R expression/NPY release and, in neuroblastoma cells, the expression of both NPY/Y5R has been positively correlated with the expressions of BDNF and its tropomyosin-related kinase B (TrkB) receptor [10,185]. A high TrkB expression correlates with a worse prognosis in patients with neuroblastoma [10]. BDNF promoted the internalization of Y5R, and Y5R antagonists blocked the pro-survival effect mediated by BDNF [185]. Moreover, under pro-apoptotic conditions, NPY/Y5R was upregulated in neuroblastoma cells (in a BDNF-independent manner) [185]. An increased Y5R expression has been reported when neuroblastomas were treated with chemotherapy, and the blockade of Y5R promoted apoptosis in tumor cells and sensitized resistant neuroblastoma cells to chemotherapy [185]. Thus, the survival NPY/Y5R system is activated by BDNF, and Y5R increases the pro-survival action of this factor and mediates an additional BDNF-independent anti-apoptotic activity contributing to the chemoresistance of neuroblastoma cells. The survival effect exerted by BDNF is mediated by p44/p42 MAPK and PI3K/Akt pathways; Y5R, upon BDNF stimulation, is transactivated by TrkB, enhancing the activation of the p44/p42 MAPK pathway [10]. Therefore, Y5R interacts with other surface receptors, such as TrkB.

Serum NPY level has been suggested as a biomarker for neuroblastoma [188]. High serum NPY levels correlate with relapse, metastasis, and poor survival, whereas a high Y5R expression has been suggested as a biomarker of angio-invasive cancer cells [188]. ProNPY processing has been associated with inferior outcomes/clinically advanced stages in neuroblastoma; thus, the proNPY to NPY processing degree was lower in advanced neuroblastomas with metastatic or regional spread [189]. A poor clinical outcome was related to less than 50% of proNPY processing [189]. Moreover, Y2R expression was observed in undifferentiated cancer cells [188].

NPY increases cell survival, restores the level of neurotrophins, and counteracts the toxic action mediated by β-amyloid in neuroblastoma cells (SH-SY5Y) [190]. NPY also protects against endoplasmic reticulum stress and glutamate excitotoxicity in human neuroblastoma SH-SY5Y cells [191]. NPY increased cell viability, counteracted the glutamate-induced pro-apoptotic activation of Jnk/Bad and ERK1/2 pathways, downregulated the expressions of CHOP, phosphor-eIF2α, and BiP, blocking endoplasmic reticulum activation, and attenuated the Akt/FoxO3a pathway in acute oxidative conditions promoted by glutamate [191]. Thus, NPY increases the survival of these cells by exerting an anti-apoptotic effect. Moreover, the activation of the PKC beta isoform via the ERK1/2 pathway augmented the expression of neuronal differentiation genes in neuroblastoma cells [192] and NPY upregulated genes containing the cAMP response element (including *Y1R* gene), via cAMP response element binding protein (CREB)/intracellular Ca^++^, in SK-N-BE2 neuroblastoma cells expressing Y1R/Y2R [193]. NPY blocked Ca^++^ channel currents in human neuroblastoma cells (SH-SY5Y) [194]. NPY also regulated, through the adenylate cyclase/protein kinase A pathway, the ATP-induced increase in internal Ca^++^ in the human CHP-234 neuroblastoma cell line [195]. Moreover, NPY, via Y2R, blocked cAMP accumulation mediated by forskolin and omega-conotoxin-sensitive high K^+^-induced Ca^++^ influx and attenuated the intracellular release of Ca^++^ promoted by angiotensin II/bradykinin in human SMS-KAN neuroblastoma cells [196]. Unusually persistent Gαi-signaling of Y2R depleted cellular Gi/o pools, leading to a Gi-refractory state in neuroblastoma cells (SH-SY5Y, SMS-KAN) expressing Y2R [79]. This persistent signaling promoted a form of cellular desensitization of the inhibitory Gαi pathway; tumor cells showed a refractory state, preventing further Y2R Gαi-signaling itself but, in addition, that of other Gαi/o-coupled receptors by regulating the downstream effectors [79].

Metastasis has been associated with a high release of NPY from neuroblastoma cells; NPY promoted the motility and invasiveness of these cells, and the peptide acted as a chemotactic factor for neuroblastoma cells [68]. These effects, via the RhoA activation pathway, were mainly mediated by Y5R, leading to cytoskeleton remodeling and cell movement [68]. Y5R is highly expressed in migratory cells in neuroblastoma tissue and cultured neuroblastoma cells; hence, Y5R is a promising antimetastatic target.

### 3.16. Ovarian Cancer

Y1R and Y2R have been reported in ovarian cancer and tumor-associated blood vessels [47]. NPY exerted a protective action against cisplatin and improved bone marrow dysfunction in an ovarian tumor murine model [197]. 

PYY has been observed in the insular type of ovarian carcinoid tumor [198] and ovarian stromal carcinoid [199,200].

### 3.17. Pancreatic Cancer

NPY, Y1R, and Y2R expressions have been studied in human pancreatic ductal adenocarcinomas and a mouse model of pancreatic cancer [201]. NPY/Y1R expressions were observed in murine/human samples, but these expressions were similar in non-neoplastic/neoplastic tissues. However, in both mouse and human samples, an overexpression of Y2R was observed in pancreatic cancer compared with that found in non-neoplastic tissues [201]. 

NPY promoted the growth of exocrine pancreatic carcinoma cells [202], and the peptide was released from human insulinomas, leading to a high serum NPY level, which returned to a standard level after surgical resection of the tumors [203]. Glibenclamide (used to treat type 2 diabetes mellitus) promoted the release of NPY from hamster insulinoma tumor cells [204], and NPY blocked insulin release in the insulinoma RIN 5AH cell line; the latter action was due to the Y1R/adenylyl cyclase inhibition [205].

PYY decreased the growth of human Mia PaCa-2 and BxPC-3 pancreatic tumor cells [206,207] and cAMP levels in Mia PaCa-2 cells [207]. Moreover, in an experimental animal model, the growth of Mia PaCa-2 cells was decreased after treatment with BIM-43004-1, a modified PYY22-36 Y2R synthetic agonist; this agonist also reduced the level of cAMP in Mia PaCa-2 cells and the growth of PANC-1 pancreatic tumor cells [207,208]. PYY also exerted an antitumor action against the latter cells [208]. The growth inhibition capacity of several PYY fragments (e.g., PYY1-36, PYY14-36, PYY22-36) has been studied; PYY14-36 showed the highest antitumor potency against PANC-1 and Mia PaCa-2 pancreatic cancer cells as well as the highest specific binding to tumor cells [209]. Moreover, the co-administration of PYY and vitamin E significantly increased the antitumor action against Mia PaCa-2 pancreatic tumor cells [210]. This action was lower when each compound was administered alone. On the contrary, PYY increased the growth of Capan-2 pancreatic tumor cells, whereas the peptide did not significantly increase the growth of Mia PaCa-2 pancreatic tumor cells [202].

A low plasma level of PP has been reported in patients with pancreatic ductal adenocarcinoma [211], whereas an elevated plasma PP level was reported in patients with an endocrine pancreatic tumor [212]. PP has been reported in the pancreatic head region [213], and a blunted PP response to a mixed meal was observed in patients with pancreatic head cancer [214]. High PP levels have been reported in patients with pancreatic neuroendocrine tumors [215], and PP was released from a pancreatic neuroendocrine tumor [216]. PP has also been observed in insulinomas, and a study of protein meal-stimulated PP release showed higher plasma PP levels in patients with insulinomas than in the control group [217]. Elevated serum PP levels were observed in patients showing pancreatic tumors in multiple endocrine neoplasia type I due to the high incidence of PPomas observed in these patients [218]. Carbachol promoted PP release from human glucagonoma cells [219] as well as from human vasoactive intestinal polypeptide (VIP)oma; the latter release was blocked with the somatostatin analog SMS 201-995 [220]. 

### 3.18. Parathyroid Adenoma

The density of nerve fibers innervating human parathyroid adenomas varies considerably; thus, some tumors showed numerous nerve fibers containing NPY, whereas others showed a few scattered fibers containing the peptide [221]. The physiological significance of these findings is currently unknown.

### 3.19. Pheochromocytoma

Y1R, Y2R, and Y5R have been reported in pheochromocytoma [10]. Plasma NPY levels were higher in patients with adrenal pheochromocytoma than in healthy patients, or those with extra-adrenal pheochromocytoma, and tissue NPY levels were also higher in patients with adrenal pheochromocytoma than in those with extra-adrenal pheochromocytoma [222]. It is important to note that the whole NPY1-36 molecule was the predominant form found in both plasma and tissue of patients suffering from adrenal pheochromocytoma; however, many NPY fragment types were reported in patients with extra-adrenal pheochromocytoma [222]. Moreover, NPY mRNA expression was more frequently observed in adrenal than in extra-adrenal tumors [222]. The increased *NPY* gene expression in adrenal tumors favored NPY release from the neoplasm, increasing the plasma NPY level. Plasma NPY increased during the surgical removal of human pheochromocytoma tumors, and this level remained high until tumor resection [223]. As a result, these tumors release NPY into the vascular system. It has been suggested that the plasma NPY level could be used as an early diagnostic marker for pheochromocytomas and paragangliomas in patients who received treatments that interfere with catecholamine reuptake or suffer from severe kidney impairment [224]. Lower NPY levels were observed in pheochromocytomas from von Hippel-Lindau patients (noradrenergic phenotype) when compared with pheochromocytomas from multiple endocrine neoplasia type 2 patients (adrenergic phenotype) [225]. Moreover, NPY exerted a mitogenic action promoting left ventricular hypertrophy in patients with pheochromocytoma [226]. Plasma NPY levels were higher in pheochromocytoma patients with left ventricular hypertrophy than those without left ventricular hypertrophy [226].

The NPY mRNA level was increased when cultured human pheochromocytoma cells were treated with nerve growth factor, protein kinase modulators (Bu)(2)cAMP, or staurosporine, but it was reduced when they were treated with dexamethasone or insulin-like growth factor II [227]. Moreover, the NPY mRNA level was constant and low in the normal adrenal medulla, but its level widely varied in benign and malignant pheochromocytomas [227]. Thus, it seems that NPY expression is not associated with malignancy. NPY is released from pheochromocytomas [228], and nitric oxide favors the release of NPY from rat pheochromocytoma PC12 cells [229]. Bradykinin or angiotensin II also helped release NPY from pheochromocytoma cells [230,231], whereas dopamine blocked the release of NPY from pheochromocytoma PC12 cells [232]. NPY inhibited the synthesis of catecholamines in the latter cells [233]. 

### 3.20. Pituitary Adenoma

Protein/mRNA NPY and mRNA Y1R/Y2R expressions have been reported in human pituitary adenomas; a positive correlation was also observed between Y2R and NPY levels [234]. NPY immunoreactivity was observed in both the cytoplasm and nuclei of pituitary tumor cells; however, NPY was not detected in normal pituitary cells [235]. Moreover, NPY promoted the release of growth hormone from patients with prolactinoma, but prolactin release was not affected [236]; however, the peptide blocked the release of this hormone from human pituitary somatotropic tumors [237].

### 3.21. Prostate Cancer

A transcriptomic study (17,967 prospective samples) focused on the expression of NPY in prostate cancer development has been performed [238]. A low NPY expression has been associated with aggressive grade, higher genomic risk, and shorter metastases-free survival/progression-free survival. Moreover, the scarce NPY/ERG+ subtype was associated with the highest risk/rate for metastasis, whereas the low NPY/ERG- subtype was associated with the lowest risk/rate for metastasis [238]. Thus, a diminished NPY expression means adverse genomic features and outcomes. Another study has shown that the combined analysis of pro-NPY/ERG expression was not associated with prostate cancer-specific death following radical prostatectomy, castration-based treatment, and biochemical failure risk [239]. *ERG* gene rearrangements are frequently observed in prostate cancer [240]. Transcriptional changes in ERG rearrangement-positive prostate cancer cells promoted metabolic changes (e.g., glucose uptake increase) by activating signaling molecules such as NPY [240]. Thus, it seems that metabolic changes favor ERG rearrangement in prostate cancer.

The plasma NPY level has been suggested as a biomarker for prostate cancer [241]. The expressions of NPY and Y1R, Y2R and Y5R have been studied in benign prostate, primary prostate cancer, and primary prostate cancer bone metastases; moreover, a transwell migration study on the LNCaP primary prostate cancer cell line was carried out to elucidate the chemotactic characteristics of NPY [242]. The mentioned expressions were upregulated in pre-invasive prostate intraepithelial neoplasia, primary prostate cancer, and metastases, and the presence of the NPY system was increased in the perineural invasion and extraprostatic extension areas. Moreover, Y1R and Y5R showed a high expression in bone metastases [242]. The data mean that the NPY system is involved from early to disseminated stages in primary prostate cancer development and is involved in cancer perineural spread and invasiveness.

NPY released from nerve terminals regulates therapy resistance and oncogenesis in prostate cancer [243]. Significant findings have been reported: tumor cells favored the increase in NPY-positive nerve fibers, and the inhibition of NPY promoted the scalation of apoptosis in cancer cells, changes in energetic metabolic pathways, and decreased cell motility [243]. This study demonstrated that NF-κB acted as an NPY downstream mediator and that quantifying nerve fibers containing NPY predicted prostate cancer-specific death. Moreover, nerve fibers containing NPY have been involved in radiation therapy resistance because the apoptosis induced by radiation was decreased when prostate tumor cells were cocultured with dorsal root ganglia or nerve fibers. In patients where radiation therapy failed, positive nerve fibers containing NPY increased [243]. Because of this, the NPY neural microenvironment plays a vital role in prostate oncogenesis and therapy resistance, and the NPY system is a potential antitumor target in this disease.

NPY decreases prostate tumor cell proliferation (DU145 and LNCaP cell lines) and promotes cell proliferation in prostate PC3 tumor cells [244]. These actions were mediated by Y1R and related to a time-course of MAPK/ERK1/2 phosphorylation: transient in PC3 cells and long-lasting in DU145 cells [244]. In the latter two cell lines, the Y1R antagonist BIBP3226 inhibited the ERK1/2 phosphorylation induced by NPY; this peptide, and only in PC3 cells, decreased the forskolin-stimulated cAMP accumulation, whereas changes in the level of Ca^++^ were not observed in the three cell lines studied [245]. 

Depression promotes prostate cancer development and dysregulates the immune response [246,247]. This observation is essential since depression also favored NPY release from prostate tumor cells in an experimental animal model of depression, and the peptide recruited myeloid-derived suppressor cells to the tumor [246,247]. Moreover, the latter and cancer cells released interleukin-6, activating the STAT3 signaling pathway in tumor cells. In this context, prostate cancer patients with a higher depression score had higher CD68+ tumor-associated macrophage infiltration and interleukin-6/NPY expressions [247]. 

PYY blocked the human PC3 prostate cancer cell line’s growth and increased VEGF production in these cells [248].

### 3.22. Thymus Cancer

An immunohistochemical study has reported the expression of NPY in thymomas [249]. 

Table 1 and Figure 8, Figure 9 and Figure 10 show the main findings regarding the involvement of NPY, PYY, and PP in cancer.

## 4. The NPY Peptide Family as Cancer Biomarkers/Prognostic Factors

Many reports have suggested that the expression of peptides belonging to the NPY peptide family and their receptors could be used as biomarkers or prognostic factors in cancer (Table 1). *Y6R* gene expression has been associated with tumor development in uveal melanoma and has been suggested as a predictive biomarker [173]. *NPY* gene methylation has been meant to be used as a biomarker for metastatic colorectal cancer progression [115,116,117,118,119,120] and for head and neck cancer prognosis and risk [150]. A better relapse-free survival/overall survival has been associated with a higher *Y1R* gene expression in estrogen receptor-positive breast cancer patients [98]. In this sense, plasma Y1R level could be a biomarker for metastasis/prognosis in the latter patients because a high Y1R level was associated with lymph node metastasis, advanced stages, perineural invasion, and shorter cancer-specific survival [100,101]. A low Y1R expression has been associated with poor prognosis [159], and an increased Y5R expression has been correlated with survival and tumor growth/migration and invasion in hepatocellular carcinoma [160]. A lower proNPY processing to NPY has been associated with poor outcome/clinical advanced stages in neuroblastoma [189], and it has been suggested that NPY expression is not correlated with malignancy in pheochromocytomas [227]. High NPY expression has been associated with invasiveness in melanoma [170], and a low NPY expression with increased cell proliferation and high peritumoral mast cell infiltrates in primary cutaneous melanoma [174].

Moreover, a plasma NPY decrease in patients with colorectal or gastric carcinomas has been related to body weight loss and tumor size [114]. High serum NPY levels correlate with relapse, metastasis, and poor survival in patients with neuroblastoma, and a high Y5R expression has been suggested as a biomarker for angio-invasive cancer cells [188]. Plasma NPY level has been proposed as a biomarker for prostate cancer [241], and a low NPY expression was associated with aggressive grade, higher genomic risk, and shorter metastases-free survival/progression-free survival in prostate cancer [238]. Children with leukemia and a high plasma NPY level showed a better outcome [154]. Finally, plasma PYY level has not been related to colorectal cancer risk [125]; however, low PYY levels indicate a malignant potential of the alterations observed in tumor cells [129]. Using NPY/PYY/YRs expression/plasma levels as cancer biomarkers/prognostic factors requires more in-depth studies to confirm the findings reported in this section. This point is important and must be fully developed.

## 5. Antitumor Therapeutic Strategies

Using peptide–drug conjugates is a therapeutic strategy for diagnosing and treating tumor cells overexpressing peptide receptors [4]. Y1R-preferring NPY analogs with potential therapeutic applications are used to unravel the signaling mediated by this receptor in health and diseases (e.g., cancer, mood disorders, obesity, stress response, food intake) [250,251,252,253,254]. The development of multifunctional NPY-conjugates for selective nuclear delivery of radio lanthanides (targeted radionuclide therapy) is a promising strategy to deliver therapeutically active cargos (e.g., radiometals) into tumor cells overexpressing peptide receptors (e.g., Y1R) [255]. Novel ^68^Ga-labeled NPY, short analogs with potential applications in cancer imaging, have been developed [256]. Moreover, ^99^mTc-labeled NPY short analogs showing Y1R affinity and potential applications in cancer imaging have been reported: in vitro studies demonstrated a specific cellular uptake and a high internalization rate, whereas the IC_50_ was 73.2 nM [257]. Radioligands for Y1R imaging (e.g., ^18^F-labeled triazolyl-liked argininamides) by positron emission tomography (PET) are also used [258]. Tubugi-1 (a small cytotoxic peptide-SS-NPY disulfide-linked conjugate) has been used to treat tumor cancer cells overexpressing Y1R (e.g., HT-29 and Colo320 (colon cancer cells), PC-3 (prostate cancer cells)). The study showed that this conjugate exerted a toxic selectivity against tumor cells [259]. Moreover, NPY analogs conjugated to gold nanocages have been used as Y1R-targeted agents against tumor cells (PC3 prostate cancer cells) [260]. Cells treated exclusively with gold nanocages suffered changes in their cytoskeleton upon laser irradiation, promoting the loss of microvilli related to the migration of tumor cells [260]. The observation denotes that Y1R is an important antimigration and anti-invasive target. The selective Y1R ligand (Asn^6^, Pro^34^-NPY) has been used to treat glioma [86]. This ligand improved blood–brain barrier permeability, targeting glioma cells, and [Asn^6^, Pro^34^]-NPY nano micelles also improved the therapeutic effect of doxorubicin against these cells, leading to a high survival rate [86]. This ligand also improved the delivery of tariquidar and doxorubicin into breast cancer cells, inhibiting tumor cell growth. The co-administration of Y1R ligands with tariquidar promoted a synergistic effect and improved the therapeutic efficacy against multidrug-resistant breast cancer cells [108]. Moreover, NPY-loaded immune-stimulating complexes decreased breast tumor cell growth by promoting apoptosis [99]. NPY-targeted nano-scale bubbles were widely applied in ultrasound cavitation chemotherapy of Y1R-overexpressed breast cancer cells [261]. In vivo nano-scale bubbles-NPY with doxorubicin and under ultrasound irradiation increased survival time and tumor suppression; minimal damages were observed in the vital mouse organs. Immunotherapy (e.g., peptide vaccines and monoclonal antibodies) is also a promising strategy for cancer treatment and diagnosis [262]. Monoclonal antibodies against surface receptors (e.g., human epidermal growth factor receptor 2, CD33, CD30, CD22) attached to cytotoxic agents have also been developed as an antitumor strategy [4]. The effectiveness of the combination of immunotherapy/chemotherapy has been demonstrated since this therapeutic combination improved therapeutic responsiveness and decreased the side effects mediated by chemotherapeutic drugs. Neither chemotherapy nor radiotherapy altered serum NPY levels [263]. 

Previous data show therapeutic strategies that can be applied to diagnose and treat tumor cells overexpressing peptide receptors. However, another essential method would be using peptide receptor antagonists to treat cancer cells expressing these receptors or even administering peptides as anticancer agents. Numerous data suggest that the targeting of the NPY/YR system with YR antagonists is a promising antitumor strategy and that these antagonists could be used as broad-spectrum antitumor drugs since NPY is involved in the proliferation (e.g., NPY promotes the growth of exocrine pancreatic carcinoma cells) [202], migration and invasion of tumor cells and in angiogenesis and because different tumors express YRs. NPY favors breast tumor cell proliferation/migration and angiogenesis [93,94], whereas Y5R antagonists (e.g., CGP71683A) not only decreased the cell growth/migration of these cells but also promoted the death of breast cancer cells [97]. Y5R agonists (not Y1R or Y2R agonists) also initiated VEGF release from breast tumor cells, favoring angiogenesis [93]. Thus, an anti-angiogenic effect could be exerted using Y5R antagonists. Y2R antagonists inhibited tumor growth and NPY-induced angiogenesis in colorectal cancer [121], and the Y2R antagonist BIIE0246 decreased melanoma tumor weight, angiogenesis, and serum VEGF level [171]. NPY, released from neuroblastoma cells, favored cancer cell proliferation and the vascularization of tumors, which were counteracted with Y2R antagonists but not with Y5R antagonists [10,175,179]. However, in neuroblastoma cells, Y5R antagonists blocked the pro-survival effect mediated by BDNF (which favors NPY/Y5R expression/NPY release), promoted apoptosis, and sensitized resistant tumor cells to chemotherapy [10,185]. Y1R and Y5R are highly expressed in prostate cancer patients with bone metastases [242]. Thus, the use of Y1R/Y5R antagonists could counteract the development of metastases. NPY regulates energy homeostasis and exerts an orexinergic action; therefore, YR antagonists could interfere with both mechanisms [10]. However, Y2R expressed in the brain mediates an anorexigenic activity of NPY; Y2R antagonists would increase food intake, and the blockade of Y2R at the periphery would shift the energy balance from fat accumulation to an augmentation in lean mass [10]. Y5R antagonists decrease food intake, and weight loss occurred without clinical significance [10].

In addition to YR antagonists, peptides can exert an antitumor effect, opening the door to another important antitumor research line: employing peptides as anticancer agents. This finding applied to PYY, which blocked breast/gastric tumor cell growth [110,111,112] and the proliferation, migration, and invasion of colorectal cancer cells and promoted apoptosis in the latter cells [131]. In this context, colon cancer lines counteract the antitumor effect mediated by PYY by increasing the expression of Bcl-2, which blocks apoptotic mechanisms [133]. PYY also blocked the growth of hepatic carcinoma/prostate cancer cells and decreased tumor volume/weight [164,248], and PYY3-36 inhibited the growth of esophageal cancer cell lines by inducing apoptosis [137]. Moreover, PYY and its fragments (e.g., PYY14-36) decreased the development of pancreatic tumor cells [206,207,209], and BIM-43004-1, a modified PYY22-36 Y2R synthetic agonist, also reduced the growth of these cells [207,208]. NPY also inhibited tumor cell growth and invasion in cholangiocarcinoma; both effects were counteracted with antibodies against NPY [113], and via Y1R, NPY blocked tumor cell growth in hepatocellular carcinoma [159]. Y2R inhibitors, not Y1R/Y5R inhibitors, inhibited the tumor growth blockade mediated by NPY [113]. Moreover, NPY, via Y1R/Y5R, promoted the death of tumor cells in Ewing sarcoma [10,138,139,140], but under hypoxia, NPY acted as a tumor growth-promoting agent because these conditions favored Y2R expression and the formation of NPY3-36 that served as a Y2R/Y5R agonist and favored the migration of tumor cells [10,140]. Tumor hypoxia exacerbates bone metastasis in Ewing sarcoma, but this was counteracted with the blockade of Y5R [145]. These findings are significant since certain conditions (e.g., hypoxia) balance the growth-inhibitory pathway mediated by NPY through Y1R/Y5R and activate the Y2R/Y5R/NPY3-36 pathway, which mediates a growth-promoting action in Ewing sarcoma. Hypoxia also favored Y2R expression by intratumoral endothelial cells, favoring angiogenesis, which was blocked with Y2R antagonists; additionally, these antagonists could halt the migration capacity of cancer cells mediated by NPY3-36 [10,140]. NPY also decreased the proliferation of specific prostate tumor cells [244].

## 6. Future Research

The first important aim that must be achieved is to perform a systematic study on the expression/role of NPY, PYY, PP, and their receptors in different types of cancers. Most of the studies have been focused on NPY, followed by PYY, whereas information regarding PP is lacking. These studies are mainly descriptive since they were focused on the expression or absence of these peptides and their receptors in tumor cells or on the peptidergic plasma level found in patients with cancer. 

Studies on the antitumor actions exerted by peptides of the NPY peptide family (PYY, NPY) must be potentiated. In particular, the anticancer effect is generally mediated by PYY against tumors. Preclinical in vitro and in vivo studies must be developed to test/confirm the anticancer properties of PYY, to understand in-depth the YRs and signaling pathways involved, and to establish antitumor strategies. Moreover, YR antagonists must be fully developed since only a few studies have focused on the anticancer actions mediated by these antagonists. More antagonists must be tested, and preclinical research (in vitro and in vivo experiments) must be urgently developed; this will serve to establish potential therapeutic antitumor strategies to be applied in the future in clinical practice. Bone destruction degree was positively correlated with the level of NPY released from tumors [144]. Accordingly, an antitumor action is exerted with Y5R antagonists; however, NPY promoted, via Y1R/Y2R, an antinociceptive activity in cancer-induced bone pain [81]. This finding highlights the importance of developing specific Y5R antagonists that exert an antitumor action without affecting other YRs, such as Y1R/Y2R, which mediate the antinociceptive effect. More studies must be focused on the involvement of peptide fragments in tumor growth since peptide fragments (e.g., PYY1-36, PYY14-36, PYY22-36) do not show the same antitumor potency/specific binding in cancer cells [209]. Another crucial research direction must be focused on the tumor microenvironment since, under certain conditions such as hypoxia, low oxygen levels counteract the growth-inhibitory pathway mediated by NPY through Y1R/Y5R and activate the Y2R/Y5R/NPY3-36 pathway, which mediates a growth-promoting action in tumor cells [10,138,139,140]. The cross-talk between cancer progression and the release of peptides belonging to the NPY peptide family from nerve fibers must be fully understood since these peptides regulate tumor development [88]. NPY released from nerve terminals regulates oncogenesis and therapy resistance in prostate cancer: tumor cells favored the increase in NPY-positive nerve fibers, and the inhibition of NPY promoted apoptosis in cancer cells [243]. Thus, the NPY neural microenvironment plays a vital role in both mechanisms. It will be relevant to know how the expression of YRs is regulated to develop anticancer strategies; this research line must also be potentiated. The knowledge about the agents that control the expressions of NPY, PYY, and PP must be increased to develop new antitumor strategies. It is also important to note that social isolation augmented cancer development risk, and NPY levels were increased [106]; that chronic stress augmented the extent/frequency of bone metastases [107], and that depression promoted prostate cancer development, favored NPY release from prostate tumor cells and dysregulated the immune response [246,247]. They are three examples of the extensive work that must be developed to fully understand the involvement of NPY, PPY, and PP in cancer. 

Interaction of YRs with other receptors must also be studied in other tumors to understand how, for example, the brain-derived neurotrophic factor, via the TrkB receptor, activates the NPY/Y5R survival system and how Y5R increases the pro-survival action of this factor [10]. This result is essential since the expression of membrane proteins changes due to multidrug resistance, and multidrug-resistant tumor cells overexpress Y1R and P-glycoprotein [108]. 

Another important finding must be studied in-depth and confirmed: normal tissues mainly express Y2R, but Y1R was overexpressed in primary tumors and tumor-derived metastases [6]. Thus, a crucial change occurs in the neoplastic transformation from Y2R to Y1R expression [95]. Y2R expression was observed in undifferentiated cancer cells [188], and higher NPY levels have been reported in differentiated tumors, but these levels were lower in poorly differentiated tumors [89]. This observation must be substantiated and the physiological significance elucidated. Experimental evidence supports that Y2R promotes glycolysis in neuroblastoma cells; this metabolic pathway is crucial for cancer cells to obtain ATP, suggesting that Y2R antagonists could block this process [186]. This study must be thoroughly confirmed in other tumors and might assist in developing future anticancer strategies. A considerable role of NPY is, via Y1R and by decreasing the expression of pro-apoptotic proteins, the protection exerted against the nephrotoxicity induced by cisplatin [152]. NPY also exerted a protective action against cisplatin and improved bone marrow dysfunction in an ovarian tumor murine model [197]. This point must be well understood to develop combined antitumor strategies using peptides/YR antagonists and chemotherapy. Finally, the involvement of NPY, PYY, and PP in inflammatory mechanisms must be exhaustively investigated, since NPY, via Y1R, can force an anti-inflammatory effect or a pro-inflammatory action, inducing tumorigenesis [82,83].

## 7. Conclusions

The involvement of peptides and their receptors in cancer progression opens the door to developing anticancer strategies, such as applying peptides or receptor antagonists as anticancer drugs. Data regarding the proliferative/antiproliferative actions mediated by PP on tumor cells are currently lacking; however, PYY and its fragments exert, in general, an antitumor effect by blocking tumor cell growth, migration, and invasion. Thus, plenty of data suggest that PYY could be used as an anticancer agent. Similarly, NPY promotes or counteracts tumor development; therefore, the peptide favors tumor cell growth, migration, metastasis, and angiogenesis in some tumors, whereas, in others, it exhibits an antitumor effect. Hence, it is crucial to know the YR types that express tumor cells, the G proteins involved in these mechanisms, and the signaling pathways activated by the different YRs in order to explain cancer/anticancer actions mediated by NPY and PYY and to develop specific antitumor strategies. YR antagonists could exert an antitumor effect by inhibiting the tumor cell proliferation mediated by NPY. This anticancer strategy must be fully exploited. Thus, the use of YR antagonists (e.g., Y2R or Y5R antagonists) as broad-spectrum antitumor drugs is a promising anticancer strategy because (1) peptides belonging to the NPY peptide family are involved in the proliferation, migration, and invasion of tumor cells as well as in angiogenesis; (2) tumor cells express YRs, and (3) YR antagonists promote the death of tumor cells. Notably, the same YR antagonist could exert an antitumor action against many tumors expressing YRs. 

NPY levels have been suggested as tumor biomarkers and prognostic factors; however, this must be studied more and confirmed in several tumors. Most studies regarding the involvement of the NPY peptide family in tumor development have focused on NPY, whereas only incomplete or scarce data are currently available about the involvement of PYY/PP in many cancer types. Thus, more systematic studies must be performed to show the expression or not of PYY/PP and their receptors in different tumors. New broad-spectrum antitumor drugs and drug-design studies must be developed to understand the structure–function relationships between NPY/PYY/PP and their receptors. This knowledge will aid in developing specific anticancer strategies. Finally, tumor microenvironment conditions, the cross-talk between cancer progression and NPY/PYY/PP release from nerve fibers, the interaction between YRs and other receptors (e.g., TrkB), signaling pathways mediated by YRs in tumor proliferation, how the expression of NPY/PYY/PP and their receptors are regulated, and the antitumor strategy combination using NPY/PYY/PP/YR antagonists and chemotherapy must be studied in-depth and elucidated. 

Although the involvement of peptides in cancer is a promising line of research to develop specific antitumor therapeutic strategies, there is much to investigate regarding the participation of NPY, PYY, and PP in cancer progression: basic information on many tumors is scarce, incomplete, or absent. For example, it is fundamental to learn whether the expression of YR is essential or not for the viability of tumor cells; treatments against metastatic and angiogenic mechanisms and those increasing the chemosensitivity of cancer cells must also be studied and potentiated; the expression of NPY/PYY/PP and their receptors as biomarkers and prognostic factors in cancer must be fully confirmed; and studies focused on single nucleotide polymorphisms must be performed. This knowledge will aid in exploring therapeutic strategies targeting tumor-specific molecular derangements, which could be applied alone or in combination therapy with chemotherapy/radiotherapy. Finally, we hope this review will stimulate scientific interest in the involvement of peptides in cancer.

## Figures and Tables

**Figure 1 ijms-24-09962-f001:**
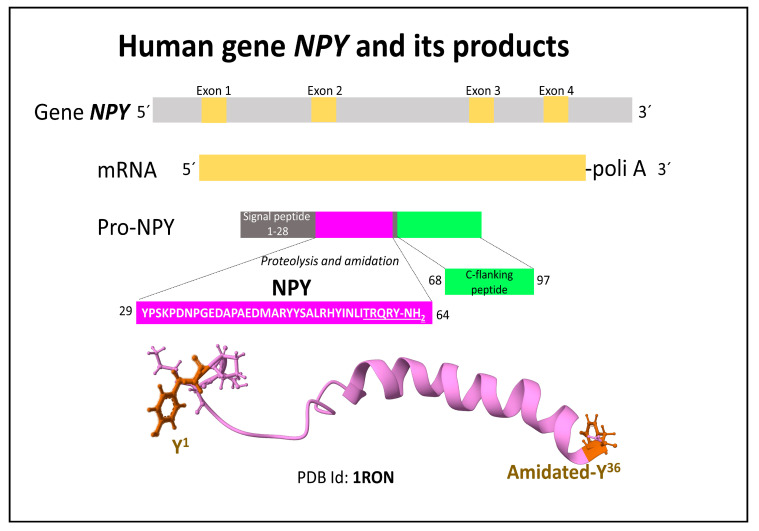
Protein products encoded by the human h*NPY* gene. The numbers indicate amino acid positions within the peptides. The NMR secondary structure of human NPY (1RON, [24]) is from the Protein Data Bank [25], drawn with the free web-based software Mol* (https://molstar.org/ (accessed on 5 May 2023)) [26]. The underlined C-terminal pentapeptide sequence is identical in all three members of the human NPY family.

**Figure 3 ijms-24-09962-f003:**
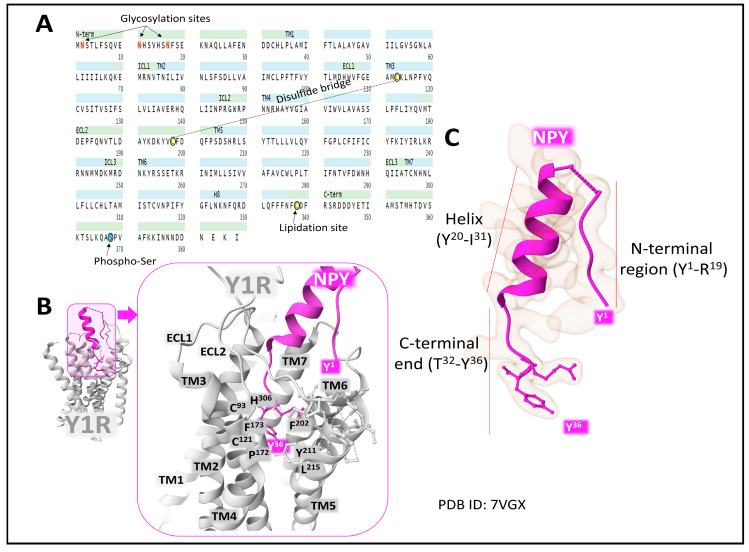
The amino acid sequence of Y1R depicting the protein domains (transmembrane, TM; extracellular loops, ECL; intracellular loops, ICL; Helix 8, H8; N-terminal and C-terminal) and the posttranslational modifications (**A**) [49]. The three-dimensional structure of Y1R bound to NPY, miniature view on the left and closer view on the right, indicating some amino acid residue interactions (**B**), and the Gaussian volume representation of NPY (**C**) [50] were obtained from the Protein Data Bank [25] and drawn with the free web-based software Mol* (https://molstar.org/ (accessed on 5 May 2023)) [26].

**Figure 4 ijms-24-09962-f004:**
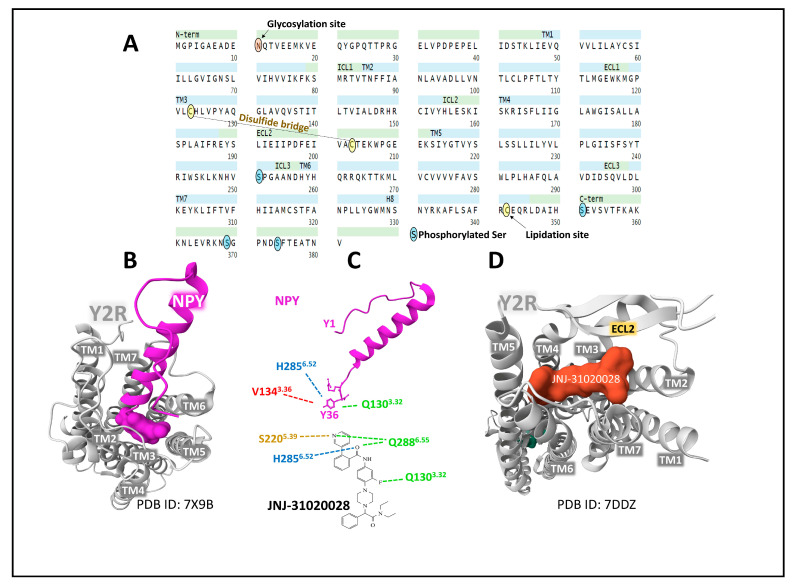
The amino acid sequence of Y2R depicting the protein domains (transmembrane, TM; extracellular loops, ECL; intracellular loops, ICL; Helix 8, H8; N-terminal and C-terminal) and the posttranslational modifications (**A**) [49]. (**B**,**D**) depict an extracellular view of the three-dimensional structure of Y2R bound to NPY (PDB ID 7X9B, [51]), indicating the Gaussian volume of the C-terminal end of NPY (**B**) and the interaction of the antagonist (molecular surface representation) with Y2R (PDSB ID 7DDZ, [51]). (**C**) shows crucial interactions between the C-terminal domain amidated tyrosine of NPY [51] and antagonist JNJ-31020028 with receptor amino acids [51]. The three-dimensional structures were obtained from the Protein Data Bank [25] and drawn with the free web-based software Mol* (https://molstar.org/ (accessed on 5 May 2023)) [26]. The structure of JNJ-31020028 was illustrated with KingDraw free software (Version 1.1.0). Superscript numbers in amino acid residues correspond to the Ballesteros–Weinstein numbering system [52].

**Figure 5 ijms-24-09962-f005:**
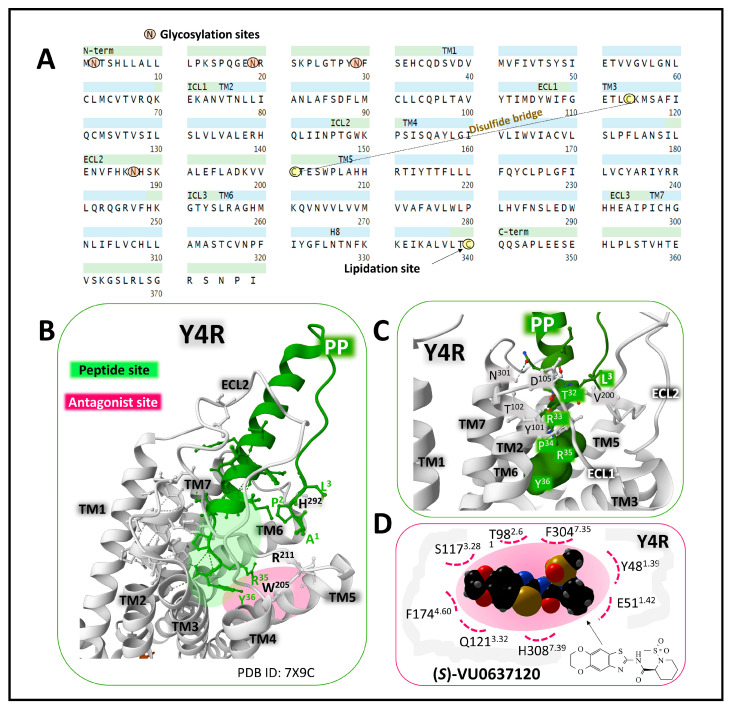
The amino acid sequence of Y4R depicting the protein domains (transmembrane, TM; extracellular loops, ECL; intracellular loops, ICL; Helix 8, H8; N-terminal and C-terminal) and the posttranslational modifications (**A**) [49]. The three-dimensional structure of Y4R bound to PP, indicating its position within the peptide site in the context of the transmembrane helices (**B**), the representation of the binding site occupied by the C-terminal end of PP (**C**), and the allosteric binding pocket for the antagonist (S)V-0637120 (3-D and 2-D structures) (**D**), drawn with KingDraw software (Version 1.1.0). (**C**,**D**) show key residues of Y4R, delimiting the orthostatic peptide site [51,55] and the antagonist allosteric binding pocket (superscripts indicate numbering according to the Ballesteros–Weinstein system to identify involved helices) [56]. The images in (**B**,**C**) were obtained from the Protein Data Bank, PDB [25], ID 7X9C [55] and drawn with the free web-based software Mol* (https://molstar.org/ (accessed on 5 May 2023)) [26].

**Figure 6 ijms-24-09962-f006:**
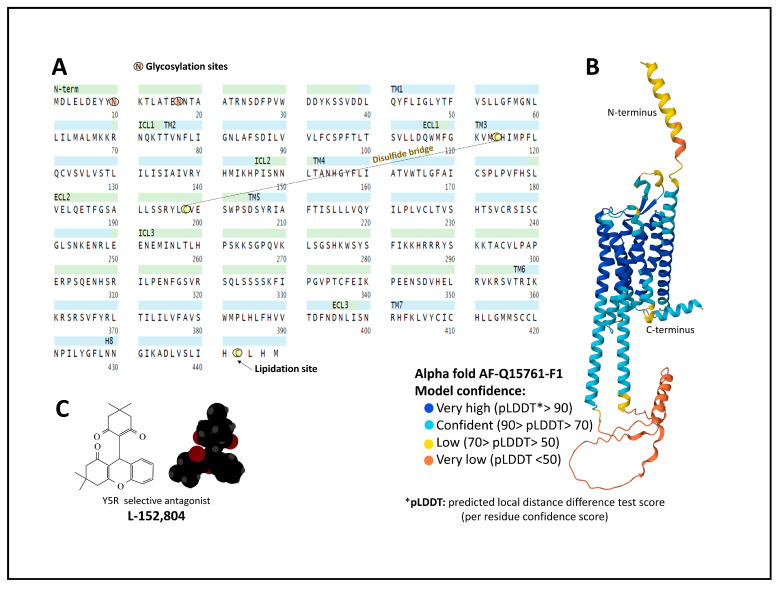
The amino acid sequence of Y5R showing the protein domains (transmembrane, TM; extracellular loops, ECL; intracellular loops, ICL; Helix 8, H8; N-terminal and C-terminal) with detailed posttranslational modifications (**A**) [49]. The predicted three-dimensional AlphaFold [61] structure of Y5R (**B**) was obtained from UniProt [22] and drawn with the free web-based software Mol* (https://molstar.org/ (accessed on 5 May 2023)) [26]. Y5R antagonist L-182,804 2D and 3D structures (**C**) were traced with KingDraw free software (Version 1.1.0).

**Figure 7 ijms-24-09962-f007:**
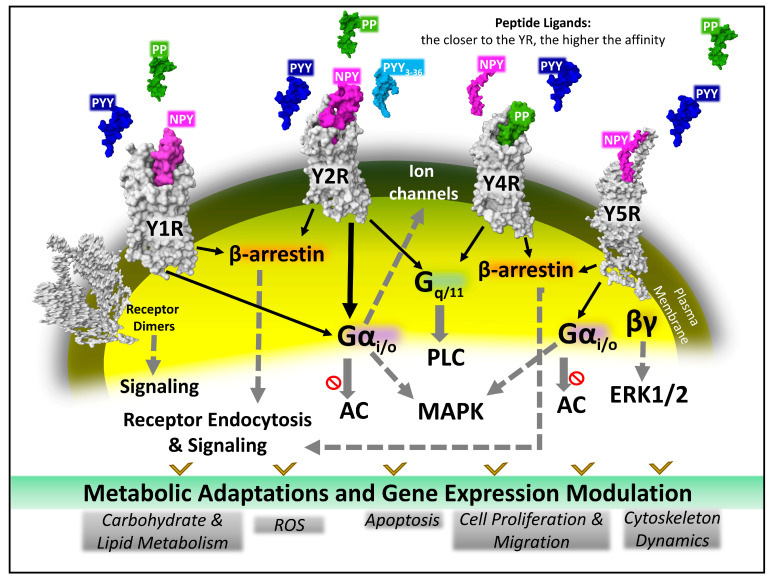
Representative primary transduction mechanisms that determine intracellular signaling pathways triggered by the binding of NPY peptides to human YRs. The black arrows indicate preferential but not exclusive transduction mechanisms [60]. Dashed gray arrows depict multiple steps taking place. Y2R exhibits a biased preference for transducer Gi over β-arrestin [53]. Receptor proteins and peptides are molecular surface representations of Protein Data Bank [25] structures 1RON (NPY), 2DEZ (PYY), 2DFO (PYY3-36), 1LJV (PP), 7VGX (Y1R), 7X9B (Y2R), 7X9C (Y4R), and AlphaFold prediction [61] for Y5R [22], illustrated with the free web-based software Mol* (https://molstar.org/, accessed on 5 May 2023) [26]. Abbreviations: AC, adenylyl cyclase; ERK1/2, extracellular receptor kinases; MAPK, mitogen-activated protein kinase; PLC, phospholipase C.

**Figure 8 ijms-24-09962-f008:**
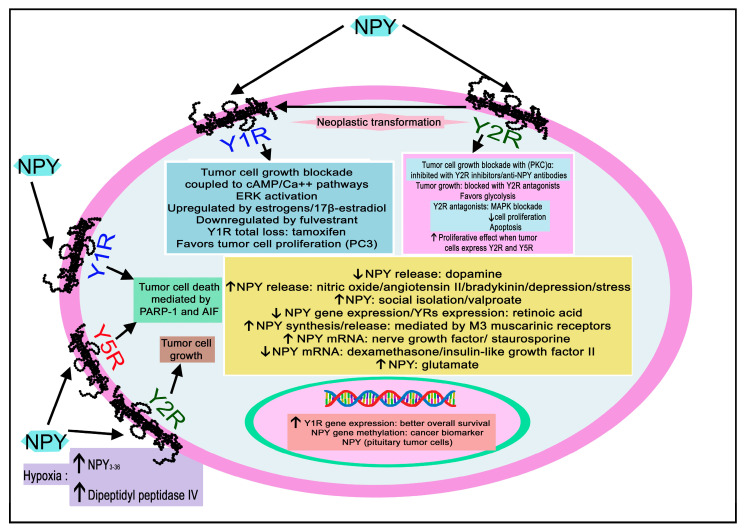
Summary of the mechanisms mediated by NPY via Y1R/Y2R in tumor cells. ↑: increase; ↓: decrease. AIF: apoptosis-inducing factor; cAMP: cyclic adenosine monophosphate; ERK: extracellular signal-regulated protein kinase; MAPK: mitogen-activated protein kinase; PARP-1: poly(ADP-ribose) polymerase; PKC: protein kinase C.

**Figure 9 ijms-24-09962-f009:**
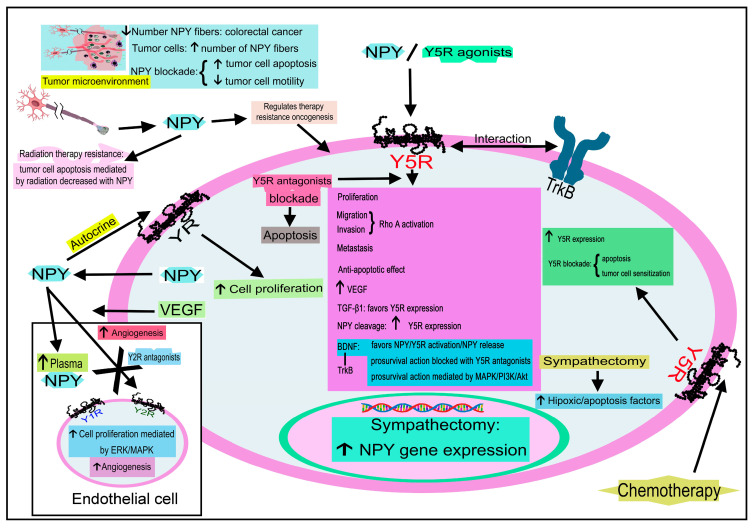
Summary of the mechanisms mediated by NPY, via Y5R, in tumor cells. ↑: increase; ↓: decrease. BDNF: brain-derived neurotrophic factor; MAPK: mitogen-activated protein kinase; PI3K: phosphatidyl-inositol-3-kinase; TGF: tumor growth factor; TrkB: tropomyosin-related kinase B receptor; VEGF: vascular endothelial growth factor.

**Figure 10 ijms-24-09962-f010:**
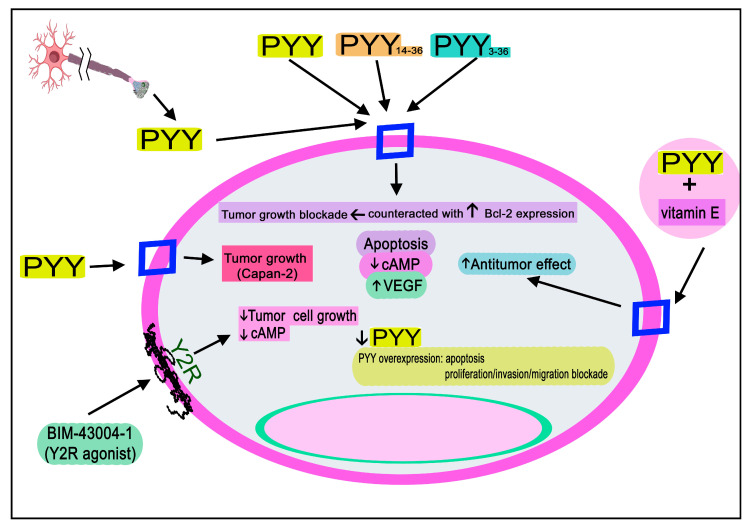
Summary of the mechanisms mediated by PYY in tumor cells. ↑: increase; ↓: decrease. cAMP: cyclic adenosine monophosphate; VEGF: vascular endothelial growth factor.

**Table 1 ijms-24-09962-t001:** NPY, PYY, and PP involvement in cancer.

Tumor	NPY-PYY-PP
Brain	Y1R high expression [86]. Y2R high expression: glioblastoma (grade IV) [88]. Medulloblastomas, meningiomas, and primitive neuroectodermal tumors: Y1R/Y2R expressions [88]. Astrocytomas (grades I to III): Y2R expression [88]. Higher NPY levels in differentiated tumors than in poorly differentiated neurological tumors [89]. Cerebellar hemangioblastoma: PYY expression [92].
Breast	Higher *Y1R* gene expression predicts a better relapse-free survival/overall survival in estrogen receptor-positive cancer patients [98].
	High Y1R level positively correlated with lymph node metastasis/clinical stage. Patients with Y1R expression: shorter cancer-specific survival [100]. High Y1R expression is associated with metastasis, advanced stages, poor Nottingham prognostic index, and perineural invasion [101]. Primary human breast tumors/breast cancer-derived metastases: Y1R overexpression [6]. Normal human breast tissues: Y2R expression [6]. Y5R mediates tumor cell growth and migration [10,94]. NPY favors proliferation/migration and angiogenesis [93,94]. NPY blocks cAMP accumulation and promotes ERK phosphorylation and tumor cell migration [97]. Y5R antagonists inhibit cell growth and migration and induce the death of tumor cells expressing Y5R [97]. Y5R agonists favor VEGF release from tumor cells, favoring angiogenesis [93]. PYY blocks tumor cell growth [110,111,112]. PYY/vitamin E co-administration: higher antitumor action [111]. PYY decreases the cAMP level [110].
Cholangiocarcinoma	Higher NPY levels in the center of tumors than in the invasion fronts [113]. NPY blocks tumor cell growth/invasion; both effects are counteracted with anti-NPY antibodies [113]. Y2R inhibitors, but not Y1R/Y5R inhibitors, inhibited tumor cell proliferation blockade mediated by NPY [113].
Colorectal	*NPY* gene methylation: a biomarker for metastatic progression [115,116,117,119,120]. Plasma NPY decrease: associated with tumor size/body weight loss [114]. NPY/Y2R overexpression. Y2R antagonists block tumor growth [121]. NPY favors angiogenesis [121]. Blood vessels considerably altered histologically [124]. Rectal tumors express PYY/PP [126]. A low PYY level indicates malignant potential [129]. PYY overexpression blocks proliferation, migration/invasion of tumor cells, favoring apoptosis [131]. Patients with colon cancer: higher PP level [134]. PP is released from tumor cells [136].
Esophageal	PYY3-36 promotes apoptosis and blocks tumor cell growth [137].
Ewing sarcoma	Plasma NPY level: high and higher in patients with tumors from a pelvic/bone origin [146]. Intratumoral blood vessels: Y1R expression [139]. NPY, Y1R, and Y5R expressions and NPY favor tumor cell death [10,138,139,140]. NPY3-36 favors tumor cell migration and angiogenesis [10,140]. Hypoxia: NPY acts as a tumor growth-promoting agent [10,140]. Hypoxia counteracts the growth-inhibitory pathway mediated by Y1R/Y5R [138,140]. Hypoxia: Y2R expression in intratumoral endothelial cells. Y2R antagonists decrease tumor vascularization [10].
	NPY/Y5R upregulated in distant metastasis. Y5R blockade counteracts bone metastasis and polyploidization [145]. Bone destruction degree: positively correlated with NPY release from tumors [144].
Gastric	Experimental model: high serum/tumor tissue PYY levels [147]. High plasma PP level [148].
Hemangioma	NPY in cells expressing Y1R [149].
Head and neck	*NPY* gene promoter methylation status: a biomarker for tumor prognosis/risk [150].
Kidney	Renal cell carcinoma: Y1R expression [151]. Nephroblastoma: Y1R/Y2R expressions [151]. Intratumoral blood vessels: high Y1R density [151]. Nerve fibers containing NPY are close to blood vessels/tumor cells [151]. Primary neuroendocrine carcinoma: PP in cancer cells [153].
Leukemia	Children with B-cell precursor leukemia: high plasma NPY level [154]. Children with high plasma NPY levels: better outcomes than those showing a normal level [154]. NPY, via Y1R, activates ERK [155]. NPY did not activate p38MAPK, jun N-terminal kinase, PKC, and phospholipase D [155]. NPY13-36 blocks intracellular Ca^++^ increase mediated by NPY [157].
Liver	NPY, via Y1R, blocks tumor cell growth [159]. Protein/mRNA Y1R level decrease [159]. Low Y1R expression: associated with poor prognosis [159]. Knockdown of Y1R: increase in tumor cell proliferation [159]. NPY/Y5R is involved in tumor cell proliferation, migration/invasion [160]. Increased Y5R expression correlated with survival and tumor growth [160]. NPY decreases PD-1+ T cells/PD-1 expression/cell in T cells and augments T cell proliferation and tumor cell eradication [162]. PYY blocks tumor cell growth, volume/weight, and cAMP level [164].
Lung	*NPY* gene expression upregulated in patients with TP53 mutation [165]. High PYY/PP levels [166,167,168,169].
Melanoma	Higher NPY expression in melanoma than in melanocytic nevi or nodular melanoma [170]. Melanoma metastases and lentigo malign melanomas: no NPY expression [170]. High NPY expression is associated with invasiveness [170]. Y2R antagonist BIIE0246: decrease in tumor weight and angiogenesis [171]. Tumors of sympathectomized animals: *NPY* gene expression and hypoxic and apoptotic factors increase [172]. Uveal melanoma: *Y6R* gene expression associated with tumor development and used as a biomarker [173]. Low NPY expression is associated with high cell proliferation [174]. High NPY expression is linked to better outcomes [174].
Neuroblastoma	High serum NPY levels correlated with relapse, metastasis, and poor survival [188]. ProNPY processing is associated with poor outcomes/clinically advanced stages [189]. NPY/Y1R/Y2R/Y4R/Y5R expression [178]. NPY favors cancer cell proliferation [10,175]. NPY increases the survival of tumor cells by exerting an anti-apoptotic effect [190,191]. NPY, via Y2R, promotes the proliferation of endothelial cells and increases vascularization; Y2R antagonists, but not Y5R antagonists, decrease vascularization [10,175,179]. Ganglioneuroblastoma: NPY expression [177]. NPY release is stimulated by protein kinase C-coupled M3 muscarinic receptors [180,181]. 17 beta-estradiol favors *Y1R* gene transcription [182]. Valproate increases NPY expression [187]. BDNF promotes NPY/Y5R expressions that are positively correlated with TrkB expression [10,185]. High TrkB expression correlated with a worse prognosis [10]. Retinoic acid decreases *NPY* gene expression and YR expression [183,184]. The Y2R antagonist BIIE0246 inhibits MAPK activation, decreases cell proliferation, promotes apoptosis, and exerts an anti-angiogenic action [10,175,179]. Y2R mediates glycolysis [186]. Chemotherapy: increase in Y5R expression [185]. Y5R blockade: promotes apoptosis in tumor cells and sensitizes resistant cancer cells to chemotherapy [185]. Metastasis is associated with a high NPY release [68]. NPY promotes motility and invasiveness in tumor cells [68]. Y5R is highly expressed in migratory cells [68].
Ovarian	Y1R/Y2R expressions [47]. NPY exerts a protective action against cisplatin [197]. PYY expression [198,199,200].
Pancreatic	NPY/Y1R expressions and Y2R overexpression [201]. NPY promotes tumor cell growth [202]. NPY released from tumors: high serum NPY level [203]. PYY decreases the growth/cAMP level in tumor cells [206,207]. BIM-43004-1 (PYY22-36 Y2R synthetic agonist): decreases tumor cell growth and cAMP level [207,208]. PYY14-36 exerts the highest antitumor action [209]. PYY/vitamin E co-administration: increase in antitumor action [210]. PYY increased the growth of Capan-2 tumor cells [202]. PP is located in the pancreatic head region and insulinoma [213]. Pancreatic ductal adenocarcinoma: low plasma PP level [211]. Endocrine pancreatic tumor: elevated plasma PP level [212]. Pancreatic neuroendocrine tumor: high PP level and peptide release [215,216]. Carbachol promotes PP release from tumor cells [219].
Parathyroid adenoma	Some tumors show numerous NPY nerve fibers, whereas others have a few scattered NPY fibers [221].
Pheochromocytoma	High plasma NPY level [222]. Plasma NPY increased during surgical tumor removal; the level remained high until tumor resection [223]. Y1R/Y2R/Y5R expressions [10]. Adrenal pheochromocytoma: NPY1-36 is the predominant molecule found in plasma/tissue [222]. Extra-adrenal pheochromocytoma: many NPY fragments [222]. NPY mRNA expression is more frequently observed in adrenal than in extra-adrenal tumors [222]. NPY mRNA level increased when tumor cells were treated with nerve growth factor, protein kinase modulators (Bu)(2)cAMP, or staurosporine, but it was reduced when treated with dexamethasone or insulin-like growth factor II [227]. NPY expression is not associated with malignancy [227]. NPY release from tumor cells: nitric oxide, bradykinin, and angiotensin II favor its release, and dopamine blocks NPY release [228,229,230,231,232]. NPY promotes left ventricular hypertrophy [226]. Plasma NPY level is higher in patients with left ventricular hypertrophy [226].
Pituitary adenoma	Protein/mRNA NPY and mRNA Y1R/Y2R expressions [234,235]. Positive correlation between Y2R and NPY level [234]. NPY favors growth hormone release and blocks prolactin release [236,237]. NPY not detected in normal pituitary cells [235].
Prostate	Low NPY expression is associated with aggressive grade, higher genomic risk, and shorter metastases-free survival/progression-free survival [238]. Transcriptional changes in ERG rearrangement-positive cancer cells promote metabolic changes by activating metabolic signaling molecules such as NPY [240]. Plasma NPY level: biomarker [241]. NPY, Y1R, Y2R and Y5R upregulation [242]. Y1R/Y5R: high expression in bone metastases [242]. NPY, released from nerve terminals, regulates therapy resistance and oncogenesis [243]. NPY blockade: increased apoptosis in cancer cells, changes in energetic metabolic pathways, and decreased cell motility [243]. NPY decreases tumor cell proliferation (DU145 and LNCaP cells) or promotes cell proliferation (PC3 cells): actions mediated by Y1R [244]. Depression favors NPY release from tumor cells, and NPY recruits myeloid-derived suppressor cells [246,247]. PYY blocks tumor cell growth and increases VEGF production in tumor cells [248].
Thymus	NPY expression [249].

## Data Availability

Not applicable.

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
