# Peer review of "Neuropeptide Y Peptide Family and Cancer: Antitumor Therapeutic Strategies"

_ijms, 2023, doi:10.3390/ijms24129962_

Round 1

Reviewer 1 Report

The authors provided a very well written review about NPY, PYY and PP receptors, their structure and functions , as well as their expression and function in cancer tissues. They also provide us with the current state of the art for drug-peptided delivery systems for cancer treatments involving such receptors. This is a very well written review and I learned a lot from reading it.

Author Response

Thanks for your positive response. We appreciate your support to highlight the crucial roles that peptides play in cancer progression.

Reviewer 2 Report

In this article, author has summarized the data available on peptide NPY, YY, PP and their receptors (YRs) and its relationship with cancer. Additionally, the roles played by these peptides in 22 different cancer types were also reviewed. Author has also described the structure and dynamics of the Neuropeptide Y Receptors. Moreover, peptide-drug conjugates strategy has been discussed for diagnosing and treating tumor cells. The review article is full of information and well written.

Major comments:

1-     Heading 3. Involvement of Neuropeptide Y, Peptide YY, and Pancreatic Polypeptide in Cancer- this section has many sub-headings if author can simplify, it will be good for readers.

2-     Table 1- should be re-formatted and references should be placed with the title.

3-     Heading 6. Future Research- should be precise and short. It will make the article better and effective, some redundant data can be removed.

4-    Heading 7. Conclusion- should be short and precise, some redundant data can be removed.

Author Response

In this article, author has summarized the data available on peptide NPY, YY, PP and their receptors (YRs) and its relationship with cancer. Additionally, the roles played by these peptides in 22 different cancer types were also reviewed. Author has also described the structure and dynamics of the Neuropeptide Y Receptors. Moreover, peptide-drug conjugates strategy has been discussed for diagnosing and treating tumor cells. The review article is full of information and well written.

Major comments:

  • Heading 3.Involvement of Neuropeptide Y, Peptide YY, and Pancreatic Polypeptide in Cancer- this section has many sub-headings if author can simplify, it will be good for readers. 

The sud-headings regarding the peptides studied (NPY, PP, PYY) have been removed. See pages 14-25.

  • Table 1-should be re-formatted and references should be placed with the title.

References have been placed after the titles. See pages 25-30.

  • Heading 6.Future Research- should be precise and short. It will make the article better and effective, some redundant data can be removed.

This section has been shortened and redundant data have been removed. The text is now more precise. See pages 35-37.

  • Heading 7. Conclusion-should be short and precise, some redundant data can be removed.

This section has been shortened and redundant data have been removed. The text is now more precise. See pages 37-38.

Round 2

Reviewer 2 Report

Accept in present form